# IMPLICIT NORMALIZING FLOWS

**Cheng Lu**[†], **Jianfei Chen**[†], **Chongxuan Li**[†], **Qiuhao Wang**[‡], **Jun Zhu**[†]*

[†]Dept. of Comp. Sci. & Tech., Institute for AI, BNRist Center
[†]Tsinghua-Bosch Joint ML Center, THBI Lab,Tsinghua University, Beijing, 100084 China
[‡]Center for Data Science, Peking University, Beijing, 100871 China
`{lucheng.lc15,chris.jianfei.chen,chongxuanli1991}@gmail.com,`
`dcszj@tsinghua.edu.cn, wqh19@pku.edu.cn`

## ABSTRACT

Normalizing flows define a probability distribution by an explicit invertible transformation $\mathbf{z} = f(\mathbf{x})$. In this work, we present implicit normalizing flows (ImpFlows), which generalize normalizing flows by allowing the mapping to be implicitly defined by the roots of an equation $F(\mathbf{z}, \mathbf{x}) = \mathbf{0}$. ImpFlows build on residual flows (ResFlows) with a proper balance between expressiveness and tractability. Through theoretical analysis, we show that the function space of ImpFlow is strictly richer than that of ResFlows. Furthermore, for any ResFlow with a fixed number of blocks, there exists some function that ResFlow has a non-negligible approximation error. However, the function is exactly representable by a single-block ImpFlow. We propose a scalable algorithm to train and draw samples from ImpFlows. Empirically, we evaluate ImpFlow on several classification and density modeling tasks, and ImpFlow outperforms ResFlow with a comparable amount of parameters on all the benchmarks.

## 1 INTRODUCTION

Normalizing flows (NFs) (Rezende & Mohamed, 2015; Dinh et al., 2014) are promising methods for density modeling. NFs define a model distribution $p_{\mathbf{x}}(\mathbf{x})$ by specifying an invertible transformation $f(\mathbf{x})$ from $\mathbf{x}$ to another random variable $\mathbf{z}$. By change-of-variable formula, the model density is

$$\ln p_{\mathbf{x}}(\mathbf{x}) = \ln p_{\mathbf{z}}(f(\mathbf{x})) + \ln |\det(J_f(\mathbf{x}))| , \tag{1}$$

where $p_{\mathbf{z}}(\mathbf{z})$ follows a simple distribution, such as Gaussian. NFs are particularly attractive due to their tractability, i.e., the model density $p_{\mathbf{x}}(\mathbf{x})$ can be directly evaluated as Eqn. (1). To achieve such tractability, NF models should satisfy two requirements: (i) the mapping between $\mathbf{x}$ and $\mathbf{z}$ is invertible; (ii) the log-determinant of the Jacobian $J_f(\mathbf{x})$ is tractable. Searching for rich model families that satisfy these tractability constraints is crucial for the advance of normalizing flow research. For the second requirement, earlier works such as inverse autoregressive flow (Kingma et al., 2016) and RealNVP (Dinh et al., 2017) restrict the model family to those with triangular Jacobian matrices.

More recently, there emerge some free-form Jacobian approaches, such as Residual Flows (ResFlows) (Behrmann et al., 2019; Chen et al., 2019). They relax the triangular Jacobian constraint by utilizing a stochastic estimator of the log-determinant, enriching the model family. However, the Lipschitz constant of each transformation block is constrained for invertibility. In general, this is not preferable because mapping a simple prior distribution to a potentially complex data distribution may require a transformation with a very large Lipschitz constant (See Fig. 3 for a 2D example). Moreover, all the aforementioned methods assume that there exists an *explicit* forward mapping $\mathbf{z} = f(\mathbf{x})$. Bijections with explicit forward mapping only covers a fraction of the broad class of invertible functions suggested by the first requirement, which may limit the model capacity.

In this paper, we propose *implicit flows* (ImpFlows) to generalize NFs, allowing the transformation to be *implicitly* defined by an equation $F(\mathbf{z}, \mathbf{x}) = \mathbf{0}$. Given $\mathbf{x}$ (or $\mathbf{z}$), the other variable can be computed by an implicit root-finding procedure $\mathbf{z} = \text{RootFind}(F(\cdot, \mathbf{x}))$. An explicit mapping $\mathbf{z} = f(\mathbf{x})$ used in prior NFs can viewed as a special case of ImpFlows in the form of $F(\mathbf{z}, \mathbf{x}) = f(\mathbf{x}) - \mathbf{z} =$

---
*Corresponding Author.

**0**. To balance between expressiveness and tractability, we present a specific from of ImpFlows, where each block is the composition of a ResFlow block and *the inverse of* another ResFlow block. We theoretically study the model capacity of ResFlows and ImpFlows in the function space. We show that the function family of single-block ImpFlows is strictly richer than that of two-block ResFlows by relaxing the Lipschitz constraints. Furthermore, for any ResFlow with a fixed number of blocks, there exists some invertible function that ResFlow has non-negligible approximation error, but ImpFlow can exactly model.

On the practical side, we develop a scalable algorithm to estimate the probability density and its gradients, and draw samples from ImpFlows. The algorithm leverages the implicit differentiation formula. Despite being more powerful, the gradient computation of ImpFlow is mostly similar with that of ResFlows, except some additional overhead on root finding. We test the effectiveness of ImpFlow on several classification and generative modeling tasks. ImpFlow outperforms ResFlow on all the benchmarks, with comparable model sizes and computational cost.

## 2 RELATED WORK

**Expressive Normalizing Flows** There are many works focusing on improving the capacity of NFs. For example, Dinh et al. (2014; 2017); Kingma & Dhariwal (2018); Ho et al. (2019); Song et al. (2019); Hoogeboom et al. (2019); De Cao et al. (2020); Durkan et al. (2019) design dedicated model architectures with tractable Jacobian. More recently, Grathwohl et al. (2019); Behrmann et al. (2019); Chen et al. (2019) propose NFs with free-form Jacobian, which approximate the determinant with stochastic estimators. In parallel with architecture design, Chen et al. (2020); Huang et al. (2020); Cornish et al. (2020); Nielsen et al. (2020) improve the capacity of NFs by operating in a higher-dimensional space. As mentioned in the introduction, all these existing works adopt *explicit* forward mappings, which is only a subset of the broad class of invertible functions. In contrast, the implicit function family we consider is richer. While we primarily discuss the implicit generalization of ResFlows (Chen et al., 2019) in this paper, the general idea of utilizing implicit invertible functions could be potentially applied to other models as well. Finally, Zhang et al. (2020) formally prove that the model capacity of ResFlows is restricted by the dimension of the residual blocks. In comparison, we study another limitation of ResFlows in terms of the bounded Lipschitz constant, and compare the function family of ResFlows and ImpFlows with a comparable depth.

**Continuous Time Flows** (CTFs) (Chen et al., 2018b; Grathwohl et al., 2019; Chen et al., 2018a) are flexible alternative to discrete time flows for generative modeling. They typically treat the invertible transformation as a dynamical system, which is approximately simulated by ordinary differential equation (ODE) solvers. In contrast, the implicit function family considered in this paper does not contain differential equations, and only requires fixed point solvers. Moreover, the theoretical guarantee is different. While CTFs typically study the universal approximation capacity under the continuous time case (i.e., "*infinite depth*" limit), we consider the model capacity of ImpFlows and ResFlows under a *finite* number of transformation steps. Finally, while CTFs are flexible, their learning is challenging due to instability (Liu et al., 2020; Massaroli et al., 2020) and exceedingly many ODE solver steps (Finlay et al., 2020), making their large-scale application still an open problem.

**Implicit Deep Learning** Utilizing implicit functions enhances the flexibility of neural networks, enabling the design of network layers in a problem-specific way. For instance, Bai et al. (2019) propose a deep equilibrium model as a compact replacement of recurrent networks; Amos & Kolter (2017) generalize each layer to solve an optimization problem; Wang et al. (2019) integrate logical reasoning into neural networks; Reshniak & Webster (2019) utilize the implicit Euler method to improve the stability of both forward and backward processes for residual blocks; and Sitzmann et al. (2020) incorporate periodic functions for representation learning. Different from these works, which consider implicit functions as a replacement to feed-forward networks, we develop *invertible* implicit functions for normalizing flows, discuss the conditions of the existence of such functions, and theoretically study the model capacity of our proposed ImpFlow in the function space.

## 3 IMPLICIT NORMALIZING FLOWS

We now present implicit normalizing flows, by starting with a brief overview of existing work.

## 3.1 NORMALIZING FLOWS

As shown in Eqn. (1), a normalizing flow $f : \mathbf{x} \mapsto \mathbf{z}$ is an invertible function that defines a probability distribution with the change-of-variable formula. The modeling capacity of normalizing flows depends on the expressiveness of the invertible function $f$. *Residual flows* (ResFlows) (Chen et al., 2019; Behrmann et al., 2019) are a particular powerful class of NFs due to their free-form Jacobian. ResFlows use $f = f_L \circ \cdots \circ f_1$ to construct the invertible mapping, where each layer $f_l$ is an invertible residual network with Lipschitz constraints bounded by a fixed constant $\kappa$:

$$f_l(\mathbf{x}) = \mathbf{x} + g_l(\mathbf{x}), \quad \mathrm{Lip}(g_l) \leq \kappa < 1, \tag{2}$$

where $\mathrm{Lip}(g)$ is the Lipschitz constant of a function $g$ (see Sec. 4.1 for details). Despite their free-form Jacobian, the model capacity of ResFlows is still limited by the Lipschitz constant of the invertible function. The Lipschitz constant of each ResFlow block $f_l$ cannot exceed 2 (Behrmann et al., 2019), so the Lipschitz constant of an $L$-block ResFlow cannot exceed $2^L$. However, to transfer a simple prior distribution to a potentially complex data distribution, the Lipschitz constant of the transformation can be required to be sufficiently large in general. Therefore, ResFlows can be undesirably deep simply to meet the Lipschitz constraints (see Fig. 3 for a 2D example). Below, we present *implicit flows* (ImpFlows) to relax the Lipschitz constraints.

## 3.2 MODEL SPECIFICATION

In general, an *implicit flow* (ImpFlow) is defined as an invertible mapping between random variables $\mathbf{x}$ and $\mathbf{z}$ of dimension $d$ by finding the roots of $F(\mathbf{z}, \mathbf{x}) = \mathbf{0}$, where $F$ is a function from $\mathbb{R}^{2d}$ to $\mathbb{R}^d$. In particular, the explicit mappings $\mathbf{z} = f(\mathbf{x})$ used in prior flow instances (Chen et al., 2019; Kingma & Dhariwal, 2018) can be expressed as an implicit function in the form $F(\mathbf{z}, \mathbf{x}) = f(\mathbf{x}) - \mathbf{z} = \mathbf{0}$. While ImpFlows are a powerful family to explore, generally they are not guaranteed to satisfy the invertibility and the tractability of the log-determinant as required by NFs. In this paper, we focus on the following specific form, which achieves a good balance between expressiveness and tractability, and leave other possibilities for future studies.

**Definition 1.** Let $g_{\mathbf{z}} : \mathbb{R}^d \to \mathbb{R}^d$ and $g_{\mathbf{x}} : \mathbb{R}^d \to \mathbb{R}^d$ be two functions such that $\mathrm{Lip}(g_{\mathbf{x}}) < 1$ and $\mathrm{Lip}(g_{\mathbf{z}}) < 1$, where $\mathrm{Lip}(g)$ is the Lipschitz constant of a function $g$. A specific form of ImpFlows is defined by

$$F(\mathbf{z}, \mathbf{x}) = \mathbf{0}, \text{ where } F(\mathbf{z}, \mathbf{x}) = g_{\mathbf{x}}(\mathbf{x}) - g_{\mathbf{z}}(\mathbf{z}) + \mathbf{x} - \mathbf{z}. \tag{3}$$

The root pairs of Eqn. (3) form a subset in $\mathbb{R}^d \times \mathbb{R}^d$, which actually defines the assignment rule of a unique invertible function $f$. To see this, for any $\mathbf{x}_0$, according to Definition 1, we can construct a contraction $h_{\mathbf{x}_0}(\mathbf{z}) = F(\mathbf{z}, \mathbf{x}_0) + \mathbf{z}$ with a unique fixed point, which corresponds to a unique root (w.r.t. $\mathbf{z}$) of $F(\mathbf{z}, \mathbf{x}_0) = \mathbf{0}$, denoted by $f(\mathbf{x}_0)$. Similarly, in the reverse process, given a $\mathbf{z}_0$, the root (w.r.t. $\mathbf{x}$) of $F(\mathbf{z}_0, \mathbf{x}) = \mathbf{0}$ also exists and is unique, denoted by $f^{-1}(\mathbf{z}_0)$. These two properties are sufficient to ensure the existence and the invertibility of $f$, as summarized in the following theorem.

**Theorem 1.** *Eqn.(3) defines a unique mapping $f : \mathbb{R}^d \to \mathbb{R}^d, \mathbf{z} = f(\mathbf{x})$, and $f$ is invertible.*

See proof in Appendix A.1. Theorem 1 characterizes the validness of the ImpFlows introduced in Definition 1. In fact, a single ImpFlow is a stack of a single ResFlow and the inverse of another single ResFlow, which will be formally stated in Sec 4. We will investigate the expressiveness of the function family of the ImpFlows in Sec 4, and present a scalable algorithm to learn a deep generative model built upon ImpFlows in Sec. 5.

## 4 EXPRESSIVENESS POWER

We first present some preliminaries on Lipschitz continuous functions in Sec. 4.1 and then formally study the expressiveness power of ImpFlows, especially in comparison to ResFlows. In particular, we prove that the function space of ImpFlows is strictly richer than that of ResFlows in Sec. 4.2 (see an illustration in Fig. 1 (a)). Furthermore, for any ResFlow with a fixed number of blocks, there exists some function that ResFlow has a non-negligible approximation error. However, the function is exactly representable by a single-block ImpFlow. The results are illustrated in Fig. 1 (b) and formally presented in Sec. 4.3.

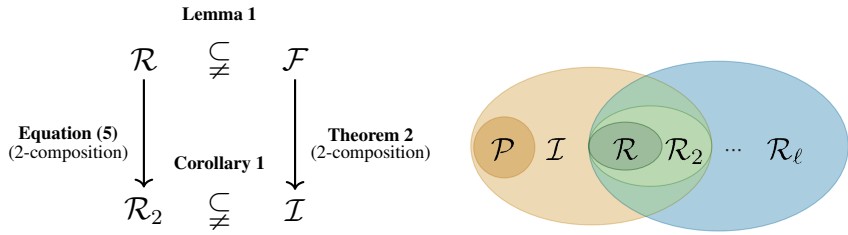

(a) Relationship between $\mathcal{R}_2$ and $\mathcal{I}$.  (b) Relationship between $\mathcal{R}_\ell$ and $\mathcal{I}$.

Figure 1: An illustration of our main theoretical results on the expressiveness power of ImpFlows and ResFlows. Panel (a) and Panel (b) correspond to results in Sec. 4.2 and Sec. 4.3 respectively.

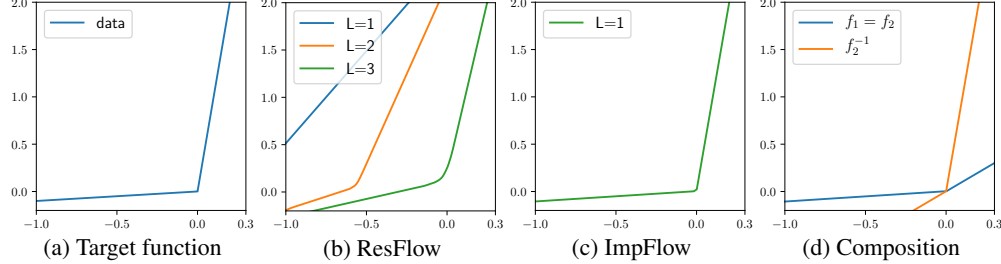

(a) Target function  (b) ResFlow  (c) ImpFlow  (d) Composition

Figure 2: A 1-D motivating example. (a) Plot of the target function. (b) Results of fitting the target function using ResFlows with different number of blocks. All functions have non-negligible approximation error due to the Lipschtiz constraint. (c) An ImpFlow that can exactly represent the target function. (d) A visualization of compositing a ResFlow block and the inverse of another ResFlow block to construct an ImpFlow block. The detailed settings can be found in Appendix D.

## 4.1 LIPSCHITZ CONTINUOUS FUNCTIONS

For any differentiable function $f : \mathbb{R}^d \to \mathbb{R}^d$ and any $\mathbf{x} \in \mathbb{R}^d$, we denote the Jacobian matrix of $f$ at $\mathbf{x}$ as $J_f(\mathbf{x}) \in \mathbb{R}^{d \times d}$.

**Definition 2.** A function $\mathbb{R}^d \to \mathbb{R}^d$ is called *Lipschitz continuous* if there exists a constant $L$, s.t.

$$\|f(\mathbf{x}_1) - f(\mathbf{x}_2)\| \leq L\|\mathbf{x}_1 - \mathbf{x}_2\|, \ \forall \mathbf{x}_1, \mathbf{x}_2 \in \mathbb{R}^d.$$

The smallest $L$ that satisfies the inequality is called the *Lipschitz constant* of $f$, denoted as $\mathrm{Lip}(f)$.

Generally, the definition of $\mathrm{Lip}(f)$ depends on the choice of the norm $\|\cdot\|$, while we use $L_2$-norm by default in this paper for simplicity.

**Definition 3.** A function $\mathbb{R}^d \to \mathbb{R}^d$ is called *bi-Lipschitz continuous* if it is Lipschitz continuous and has an inverse mapping $f^{-1}$ which is also Lipschitz continuous.

It is useful to consider an equivalent definition of the Lipschitz constant in our following analysis.

**Proposition 1.** *(Rademacher (Federer (1969), Theorem 3.1.6)) If $f : \mathbb{R}^d \to \mathbb{R}^d$ is Lipschitz continuous, then $f$ is differentiable almost everywhere, and*

$$\mathrm{Lip}(f) = \sup_{\mathbf{x} \in \mathbb{R}^d} \|J_f(\mathbf{x})\|_2,$$

*where $\|M\|_2 = \sup_{\{\mathbf{v} : \|\mathbf{v}\|_2 = 1\}} \|M\mathbf{v}\|_2$ is the operator norm of the matrix $M \in \mathbb{R}^{d \times d}$.*

## 4.2 COMPARISON TO TWO-BLOCK RESFLOWS

We formally compare the expressive power of a single-block ImpFlow and a two-block ResFlow. We highlight the structure of the theoretical results in this subsection in Fig. 1 (a) and present a 1D motivating example in Fig. 2. All the proofs can be found in Appendix. A.

On the one hand, according to the definition of ResFlow, the function family of the single-block ResFlow is

$$\mathcal{R} := \{f : f = g + \mathrm{Id}, \ g \in C^1(\mathbb{R}^d, \mathbb{R}^d), \mathrm{Lip}(g) < 1\}, \tag{4}$$

where $C^1(\mathbb{R}^d, \mathbb{R}^d)$ consists of all functions from $\mathbb{R}^d$ to $\mathbb{R}^d$ with continuous derivatives and Id denotes the identity map. Besides, the function family of $\ell$-block ResFlows is defined by composition:

$$\mathcal{R}_\ell := \{f : f = f_\ell \circ \cdots \circ f_1 \text{ for some } f_1, \cdots, f_\ell \in \mathcal{R}\}. \tag{5}$$

By definition of Eqn. (4) and Eqn. (5), $\mathcal{R}_1 = \mathcal{R}$.

On the other hand, according to the definition of the ImpFlow in Eqn. (3), we can obtain $(g_\mathbf{x} + \mathrm{Id})(\mathbf{x}) = g_\mathbf{x}(\mathbf{x}) + \mathbf{x} = g_\mathbf{z}(\mathbf{z}) + \mathbf{z} = (g_\mathbf{z} + \mathrm{Id})(\mathbf{z})$, where $\circ$ denotes the composition of functions. Equivalently, we have $\mathbf{z} = \left((g_\mathbf{z} + \mathrm{Id})^{-1} \circ (g_\mathbf{x} + \mathrm{Id})\right)(\mathbf{x})$, which implies the function family of the single-block ImpFlow is

$$\mathcal{I} = \{f : f = f_2^{-1} \circ f_1 \text{ for some } f_1, f_2 \in \mathcal{R}\}. \tag{6}$$

Intuitively, a single-block ImpFlow can be interpreted as the composition of a ResFlow block and the inverse function of another ResFlow block, which may not have an explicit form (see Fig. 2 (c) and (d) for a 1D example). Therefore, it is natural to investigate the relationship between $\mathcal{I}$ and $\mathcal{R}_2$. Before that, we first introduce a family of "monotonically increasing functions" that does not have an explicit Lipschitz constraint, and show that it is strictly larger than $\mathcal{R}$.

**Lemma 1.**

$$\mathcal{R} \subsetneq \mathcal{F} := \{f \in \mathcal{D} : \inf_{\mathbf{x} \in \mathbb{R}^d, \mathbf{v} \in \mathbb{R}^d, \|\mathbf{v}\|_2 = 1} \mathbf{v}^T J_f(\mathbf{x}) \mathbf{v} > 0\}, \tag{7}$$

*where $\mathcal{D}$ is the set of all bi-Lipschitz $C^1$-diffeomorphisms from $\mathbb{R}^d$ to $\mathbb{R}^d$, and $A \subsetneq B$ means $A$ is a proper subset of $B$.*

Note that it follows from Behrmann et al. (2019, Lemma 2) that all functions in $\mathcal{R}$ are bi-Lipschitz, so $\mathcal{R} \subsetneq \mathcal{D}$. In the 1D input case, we can get $\mathcal{R} = \{f \in C^1(\mathbb{R}) : \inf_{x \in \mathbb{R}} f'(x) > 0, \sup_{x \in \mathbb{R}} f'(x) < 2\}$, and $\mathcal{F} = \{f \in C^1(\mathbb{R}) : \inf_{x \in \mathbb{R}} f'(x) > 0\}$. In the high dimensional cases, $\mathcal{R}$ and $\mathcal{F}$ are hard to illustrate. Nevertheless, the Lipschitz constants of the functions in $\mathcal{R}$ is less than 2 (Behrmann et al., 2019), but those of the functions in $\mathcal{F}$ can be arbitrarily large. Based on Lemma 1, we prove that the function family of ImpFlows $\mathcal{I}$ consists of the compositions of two functions in $\mathcal{F}$, and therefore is a strictly larger than $\mathcal{R}_2$, as summarized in the following theorem.

**Theorem 2.** *(Equivalent form of the function family of a single-block ImpFlow).*

$$\mathcal{I} = \mathcal{F}_2 := \{f : f = f_2 \circ f_1 \text{ for some } f_1, f_2 \in \mathcal{F}\}. \tag{8}$$

Note that the identity mapping $\mathrm{Id} \in \mathcal{F}$, and it is easy to get $\mathcal{F} \subset \mathcal{I}$. Thus, the Lipschitz constant of a single ImpFlow (and its reverse) can be arbitrarily large. Because $\mathcal{R} \subsetneq \mathcal{F}$ and there exists some functions in $\mathcal{I} \setminus \mathcal{R}_2$ (see a constructed example in Sec. 4.3), we can get the following corollary.

**Corollary 1.** $\mathcal{R} \subsetneq \mathcal{R}_2 \subsetneq \mathcal{F}_2 = \mathcal{I}$.

The results on the 1D example in Fig. 2 (b) and (c) accord with Corollary 1. Besides, Corollary 1 can be generalized to the cases with $2\ell$-block ResFlows and $\ell$-block ImpFlows, which strongly motivates the usage of implicit layers in normalizing flows.

## 4.3 COMPARISON WITH MULTI-BLOCK RESFLOWS

We further investigate the relationship between $\mathcal{R}_\ell$ for $\ell > 2$ and $\mathcal{I}$, as illustrated in Fig. 1 (b). For a fixed $\ell$, the Lipschitz constant of functions in $\mathcal{R}_\ell$ is still bounded, and there exist infinite functions that are not in $\mathcal{R}_\ell$ but in $\mathcal{I}$. We construct one such function family: for any $L, r \in \mathbb{R}^+$, define

$$\mathcal{P}(L, r) = \{f : f \in \mathcal{F}, \exists \, \mathcal{B}_r \subset \mathbb{R}^d, \forall \mathbf{x}, \mathbf{y} \in \mathcal{B}_r, \|f(\mathbf{x}) - f(\mathbf{y})\|_2 \geq L \|\mathbf{x} - \mathbf{y}\|_2\}, \tag{9}$$

where $\mathcal{B}_r$ is an $d$-dimensional ball with radius of $r$. Obviously, $\mathcal{P}(L, r)$ is an infinite set. Below, we will show that $\forall \, 0 < \ell < \log_2(L)$, $\mathcal{R}_\ell$ has a non-negligible approximation error for functions in $\mathcal{P}(L, r)$. However, they are exactly representable by functions in $\mathcal{I}$.

**Theorem 3.** *Given $L > 0$ and $r > 0$, we have*

- $\mathcal{P}(L, r) \subset \mathcal{I}$.

- $\forall\, 0 < \ell < \log_2(L)$, $\mathcal{P}(L, r) \cap \mathcal{R}_\ell = \varnothing$. *Moreover, for any $f \in \mathcal{P}(L, r)$ with d-dimensional ball $\mathcal{B}_r$, the minimal error for fitting $f$ in $\mathcal{B}_r$ by functions in $\mathcal{R}_\ell$ satisfies*

$$\inf_{g \in \mathcal{R}_\ell} \sup_{\mathbf{x} \in \mathcal{B}_r} \|f(\mathbf{x}) - g(\mathbf{x})\|_2 \geq \frac{r}{2}(L - 2^\ell) \tag{10}$$

It follows Theorem 3 that to model $f \in \mathcal{P}(L, r)$, we need only a single-block ImpFlow but at least a $\log_2(L)$-block ResFlow. In Fig. 2 (b), we show a 1D case where a 3-block ResFlow cannot fit a function that is exactly representable by a single-block ImpFlow. In addition, we also prove some other properties of ImpFlows. In particular, $\mathcal{R}_3 \not\subset \mathcal{I}$. We formally present the results in Appendix B.

## 5 GENERATIVE MODELING WITH IMPFLOWS

ImpFlows can be parameterized by neural networks and stacked to form a deep generative model to model high-dimensional data distributions. We develop a scalable algorithm to perform inference, sampling and learning in such models. For simplicity, we focus on a single-block during derivation.

Formally, a parametric ImpFlow block $\mathbf{z} = f(\mathbf{x}; \theta)$ is defined by

$$F(\mathbf{z}, \mathbf{x}; \theta) = 0, \text{ where } F(\mathbf{z}, \mathbf{x}; \theta) = g_{\mathbf{x}}(\mathbf{x}; \theta) - g_{\mathbf{z}}(\mathbf{z}; \theta) + \mathbf{x} - \mathbf{z}, \tag{11}$$

and $\mathrm{Lip}(g_{\mathbf{x}}) < 1$, $\mathrm{Lip}(g_{\mathbf{z}}) < 1$. Let $\theta$ denote all the parameters in $g_{\mathbf{x}}$ and $g_{\mathbf{z}}$ (which does NOT mean $g_{\mathbf{x}}$ and $g_{\mathbf{z}}$ share parameters). Note that $\mathbf{x}$ refers to the input of the layer, not the input data.

The *inference* process to compute $\mathbf{z}$ given $\mathbf{x}$ in a single ImpFlow block is solved by finding the root of $F(\mathbf{z}, \mathbf{x}; \theta) = 0$ w.r.t. $\mathbf{z}$, which cannot be explicitly computed because of the implicit formulation. Instead, we adopt a quasi-Newton method (i.e. Broyden's method (Broyden, 1965)) to solve this problem iteratively, as follows:

$$\mathbf{z}^{[i+1]} = \mathbf{z}^{[i]} - \alpha B F(\mathbf{z}^{[i]}, \mathbf{x}; \theta), \text{ for } i = 0, 1, \cdots, \tag{12}$$

where $B$ is a low-rank approximation of the Jacobian inverse[1] and $\alpha$ is the step size which we use line search method to dynamically compute. The stop criterion is $\|F(\mathbf{z}^{[i]}, \mathbf{x}; \theta)\|_2 < \epsilon_f$, where $\epsilon_f$ is a hyperparameter that balances the computation time and precision. As Theorem 1 guarantees the existence and uniqueness of the root, the convergence of the Broyden's method is also guaranteed, which is typically faster than a linear rate.

Another inference problem is to estimate the log-likelihood. Assume that $\mathbf{z} \sim p(\mathbf{z})$ where $p(\mathbf{z})$ is a simple prior distribution (e.g. standard Gaussian). The log-likelihood of $\mathbf{x}$ can be written by

$$\ln p(\mathbf{x}) = \ln p(\mathbf{z}) + \ln \det(I + J_{g_{\mathbf{x}}}(\mathbf{x})) - \ln \det(I + J_{g_{\mathbf{z}}}(\mathbf{z})), \tag{13}$$

where $J_f(\mathbf{x})$ denotes the Jacobian matrix of a function $f$ at $\mathbf{x}$. See Appendix. A.4 for the detailed derivation. Exact calculation of the log-determinant term requires $\mathcal{O}(d^3)$ time cost and is hard to scale up to high-dimensional data. Instead, we propose the following unbiased estimator of $\ln p(\mathbf{x})$ using the same technique in Chen et al. (2019) with Skilling-Hutchinson trace estimator (Skilling, 1989; Hutchinson, 1989):

$$\ln p(\mathbf{x}) = \ln p(\mathbf{z}) + \mathbb{E}_{n \sim p(N), \mathbf{v} \sim \mathcal{N}(0, I)} \left[ \sum_{k=1}^{n} \frac{(-1)^{k+1}}{k} \frac{\left(\mathbf{v}^T[J_{g_{\mathbf{x}}}(\mathbf{x})^k]\mathbf{v} - \mathbf{v}^T[J_{g_{\mathbf{z}}}(\mathbf{z})^k]\mathbf{v}\right)}{\mathbb{P}(N \geq k)} \right], \tag{14}$$

where $p(N)$ is a distribution supported over the positive integers.

The *sampling* process to compute $\mathbf{x}$ given $\mathbf{z}$ can also be solved by the Broyden's method, and the hyperparameters are shared with the inference process.

In the *learning* process, we perform stochastic gradient descent to minimize the negative log-likelihood of the data, denoted as $\mathcal{L}$. For efficiency, we estimate the gradient w.r.t. the model parameters in the backpropagation manner. According to the chain rule and the additivity of the log-determinant, in each layer we need to estimate the gradients w.r.t. $\mathbf{x}$ and $\theta$ of Eqn. (13). In particular, the gradients computation involves two terms: one is $\frac{\partial}{\partial(\cdot)} \ln \det(I + J_g(\mathbf{x}; \theta))$ and the

---

[1] We refer readers to Broyden (1965) for the calculation details for $B$.

Table 1: Classification error rate (%) on test set of vanilla ResNet, ResFlow and ImpFlow of ResNet-18 architecture, with varying Lipschitz coefficients $c$.

|  |  | Vanilla | $c = 0.99$ | $c = 0.9$ | $c = 0.8$ | $c = 0.7$ | $c = 0.6$ |
|---|---|---|---|---|---|---|---|
| CIFAR10 | ResFlow | 6.61($\pm$0.02) | 8.24 ($\pm$0.03) | 8.39 ($\pm$0.01) | 8.69 ($\pm$0.03) | 9.25 ($\pm$0.02) | 9.94 ($\pm$0.02) |
|  | ImpFlow |  | **7.29** ($\pm$0.03) | **7.41** ($\pm$0.03) | **7.94** ($\pm$0.06) | **8.44** ($\pm$0.04) | **9.22** ($\pm$0.02) |
| CIFAR100 | ResFlow | 27.83($\pm$0.03) | 31.02 ($\pm$0.05) | 31.88 ($\pm$0.02) | 32.21 ($\pm$0.03) | 33.58 ($\pm$0.02) | 34.48 ($\pm$0.03) |
|  | ImpFlow |  | **29.06** ($\pm$0.03) | **30.47** ($\pm$0.03) | **31.40** ($\pm$0.03) | **32.64** ($\pm$0.01) | **34.17** ($\pm$0.02) |

other is $\frac{\partial \mathcal{L}}{\partial \mathbf{z}} \frac{\partial \mathbf{z}}{\partial (\cdot)}$, where $g$ is a function satisfying $\mathrm{Lip}(g) < 1$ and $(\cdot)$ denotes $\mathbf{x}$ or $\theta$. On the one hand, for the log-determinant term, we can use the same technique as Chen et al. (2019), and obtain an unbiased gradient estimator as follows.

$$\frac{\partial \ln \det(I + J_g(\mathbf{x};\theta))}{\partial(\cdot)} = \mathbb{E}_{n \sim p(N), \mathbf{v} \sim \mathcal{N}(0,I)} \left[ \left( \sum_{k=0}^{n} \frac{(-1)^k}{\mathbb{P}(N \geq k)} \mathbf{v}^T J_g(\mathbf{x};\theta)^k \right) \frac{\partial J_g(\mathbf{x};\theta)}{\partial(\cdot)} \mathbf{v} \right], \quad (15)$$

where $p(N)$ is a distribution supported over the positive integers. On the other hand, $\frac{\partial \mathcal{L}}{\partial \mathbf{z}} \frac{\partial \mathbf{z}}{\partial (\cdot)}$ can be computed according to the *implicit function theorem* as follows (See details in Appendix A.5):

$$\frac{\partial \mathcal{L}}{\partial \mathbf{z}} \frac{\partial \mathbf{z}}{\partial(\cdot)} = \frac{\partial \mathcal{L}}{\partial \mathbf{z}} J_G^{-1}(\mathbf{z}) \frac{\partial F(\mathbf{z}, \mathbf{x};\theta)}{\partial(\cdot)}, \text{ where } G(\mathbf{z};\theta) = g_{\mathbf{z}}(\mathbf{z};\theta) + \mathbf{z}. \quad (16)$$

In comparision to directly calculate the gradient through the quasi-Newton iterations of the forward pass, the implicit gradient above is simple and memory-efficient, treating the root solvers as a black-box. Following Bai et al. (2019), we compute $\frac{\partial \mathcal{L}}{\partial \mathbf{z}} J_G^{-1}(\mathbf{z})$ by solving a linear system iteratively, as detailed in Appendix C.1. The training algorithm is formally presented in Appendix C.4.

## 6 EXPERIMENTS

We demonstrate the model capacity of ImpFlows on the classification and density modeling tasks[2]. In all experiments, we use spectral normalization (Miyato et al., 2018) to enforce the Lipschitz constrants, where the Lipschitz constant upper bound of each layer (called Lipschitz coefficient) is denoted as $c$. For the Broyden's method, we use $\epsilon_f = 10^{-6}$ and $\epsilon_b = 10^{-10}$ for training and testing to numerically ensure the invertibility and the stability during training. Please see other detailed settings including the method of estimating the log-determinant, the network architecture, learning rate, batch size, and so on in Appendix D.

### 6.1 VERIFYING CAPACITY ON CLASSIFICATION

We first empirically compare ResFlows and ImpFlows on classification tasks. Compared with generative modeling, classification is a more direct measure of the richness of the functional family, because it isolates the function fitting from generative modeling subtleties, such as log-determinant estimation. We train both models in the same settings on CIFAR10 and CIFAR100 (Krizhevsky & Hinton, 2009). Specifically, we use an architecture similar to ResNet-18 (He et al., 2016). Overall, the amount of parameters of ResNet-18 with vanilla ResBlocks, ResFlows and ImpFlows are the same of 6.5M. The detailed network structure can be found in Appendix D. The classification results are shown in Table 1. To see the impact of the Lipschitz constraints, we vary the Lipschitz coefficient $c$ to show the difference between ResFlows and ImpFlows under the condition of a fixed Lipschitz upper bound. Given different values of $c$, the classification results of ImpFlows are consistently better than those of ResFlows. These results empirically validate Corollary 1, which claims that the

---

[2]See https://github.com/thu-ml/implicit-normalizing-flows for details.

Table 2: Average test log-likelihood (in nats) of tabular datasets. Higher is better.

|  | POWER | GAS | HEPMASS | MINIBOONE | BSDS300 |
|---|---|---|---|---|---|
| RealNVP (Dinh et al., 2017) | 0.17 | 8.33 | -18.71 | -13.55 | 153.28 |
| FFJORD (Grathwohl et al., 2019) | 0.46 | 8.59 | -14.92 | -10.43 | 157.40 |
| MAF (Papamakarios et al., 2017) | 0.24 | 10.08 | -17.70 | -11.75 | 155.69 |
| NAF (Huang et al., 2018) | **0.62** | 11.96 | -15.09 | **-8.86** | **157.73** |
| ImpFlow ($L = 20$) | 0.61 | **12.11** | **-13.95** | -13.32 | 155.68 |
| ResFlow ($L = 10$) | 0.26 | 6.20 | -18.91 | -21.81 | 104.63 |
| ImpFlow ($L = 5$) | **0.30** | **6.94** | **-18.52** | **-21.50** | **113.72** |

Table 3: Average bits per dimension of ResFlow and ImpFlow on CIFAR10, with varying Lipschitz coefficients $c$. Lower is better.

|  | $c = 0.9$ | $c = 0.8$ | $c = 0.7$ | $c = 0.6$ |
|---|---|---|---|---|
| ResFlow ($L = 12$) | 3.469($\pm$0.0004) | 3.533($\pm$0.0002) | 3.627($\pm$0.0004) | 3.820($\pm$0.0003) |
| ImpFlow ($L = 6$) | **3.452**($\pm$0.0003) | **3.511**($\pm$0.0002) | **3.607**($\pm$0.0003) | **3.814**($\pm$0.0005) |

functional family of ImpFlows is richer than ResFlows. Besides, for a large Lipschitz constant upper bound $c$, ImpFlow blocks are comparable with the vanilla ResBlocks in terms of classification.

## 6.2 Density Modeling On 2D Toy Data

For the density modeling tasks, we first evaluate ImpFlows on the *Checkerboard* data whose density is multi-modal, as shown in Fig. 3 (a). For fairness, we follow the same experiment settings as Chen et al. (2019) (which are specified in Appendix D), except that we adopt a Sine (Sitzmann et al., 2020) activation function for all models. We note that the data distribution has a bounded support while we want to fit a transformation $f$ mapping it to the standard Gaussian dis-

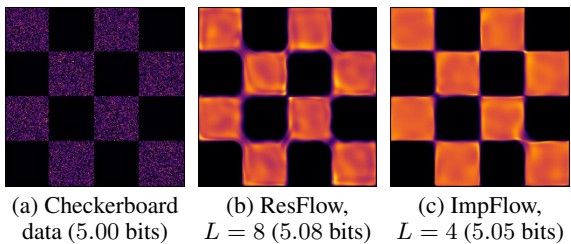

(a) Checkerboard data (5.00 bits)    (b) ResFlow, $L = 8$ (5.08 bits)    (c) ImpFlow, $L = 4$ (5.05 bits)

Figure 3: Checkerboard data density and the results of a 8-block ResFlow and a 4-block ImpFlow.

tribution, whose support is unbounded. A perfect $f$ requires a sufficiently large $\|J_f(\mathbf{x})\|_2$ for some $x$ mapped far from the mean of the Gaussian. Therefore, the Lipschtiz constant of such $f$ is too large to be fitted by a ResFlow with 8 blocks (See Fig. 3 (b)). A 4-block ImpFlow can achieve a result of $5.05$ bits, which outperforms the $5.08$ bits of a 8-block ResFlow with the same number of parameters. Such results accord with our theoretical results in Theorem 2 and strongly motivate ImpFlows.

## 6.3 Density Modeling On Real Data

We also train ImpFlows on some real density modeling datasets, including the tabular datasets (used by Papamakarios et al. (2017)), CIFAR10 and 5-bit $64 \times 64$ CelebA (Kingma & Dhariwal, 2018). For all the real datasets, we use the scalable algorithm proposed in Sec. 5.

We test our performance on five tabular datasets: POWER ($d = 6$), GAS ($d = 8$), HEPMASS ($d = 21$), MINIBOONE ($d = 43$) and BSDS300 ($d = 63$) from the UCI repository (Dua & Graff, 2017), where $d$ is the data dimension. For a fair comparison, on each dataset we use a 10-block ResFlow and a 5-block ImpFlow with the same amount of parameters, and a 20-block ImpFlow for a better result. The detailed network architecture and hyperparameters can be found in Appendix D. Table 2 shows the average test log-likelihood for ResFlows and ImpFlows. ImpFlows achieves better density estimation performance than ResFlow consistently on all datasets. Again, the results demonstrate the effectiveness of ImpFlows.

Then we test ImpFlows on the CIFAR10 dataset. We train a multi-scale convolutional version for both ImpFlows and ResFlows, following the same settings as Chen et al. (2019) except that we use a smaller network of 5.5M parameters for both ImpFlows and ResFlows (see details in Appendix D). As shown in Table 3, Impflow achieves better results than ResFlow consistently given different values of the Lipschitz coefficient $c$. Moreover, the computation time of ImpFlow is comparable to that of ResFlow. See Appendix C.2 for detailed results. Besides, there is a trade-off between the expressiveness and the numerical optimization of ImpFlows in larger models. Based on the above experiments, we believe that advances including an lower-variance estimate of the log-determinant can benefit ImpFlows in larger models, which is left for future work.

We also train ImpFlows on the 5-bit $64 \times 64$ CelebA. For a fair comparison, we use the same settings as Chen et al. (2019). The samples from our model are shown in Appendix E.

## 7 Conclusions

We propose implicit normalizing flows (ImpFlows), which generalize normalizing flows via utilizing an implicit invertible mapping defined by the roots of the equation $F(\mathbf{z}, \mathbf{x}) = 0$. ImpFlows build on Residual Flows (ResFlows) with a good balance between tractability and expressiveness. We show that the functional family of ImpFlows is richer than that of ResFlows, particularly for modeling functions with large Lipschitz constants. Based on the implicit differentiation formula, we present a scalable algorithm to train and evaluate ImpFlows. Empirically, ImpFlows outperform ResFlows on several classification and density modeling benchmarks. Finally, while this paper mostly focuses on the implicit generalization of ResFlows, the general idea of utilizing implicit functions for NFs could be extended to a wider scope. We leave it as a future work.

## Acknowledgement

We thank Yuhao Zhou, Shuyu Cheng, Jiaming Li, Kun Xu, Fan Bao, Shihong Song and Qi'An Fu for proofreading. This work was supported by the National Key Research and Development Program of China (Nos. 2020AAA0104304), NSFC Projects (Nos. 61620106010, 62061136001, U19B2034, U181146, 62076145), Beijing NSF Project (No. JQ19016), Beijing Academy of Artificial Intelligence (BAAI), Tsinghua-Huawei Joint Research Program, Huawei Hisilicon Kirin Intelligence Engineering Development, the MindSpore team, a grant from Tsinghua Institute for Guo Qiang, Tiangong Institute for Intelligent Computing, and the NVIDIA NVAIL Program with GPU/DGX Acceleration. C. Li was supported by the fellowship of China postdoctoral Science Foundation (2020M680572), and the fellowship of China national postdoctoral program for innovative talents (BX20190172) and Shuimu Tsinghua Scholar. J. Chen was supported by Shuimu Tsinghua Scholar.

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

## A    ADDITIONAL LEMMAS AND PROOFS

### A.1    PROOF FOR THEOREM 1

*Proof.* (**Theorem 1**)

Firstly, $\forall \mathbf{x}_0 \in \mathbb{R}^d$, the mapping

$$h_{\mathbf{x}_0}(\mathbf{z}) = F(\mathbf{z}, \mathbf{x}_0) + \mathbf{z}$$

is a contrative mapping, which can be shown by Lipschitz condition of $g_z$ :

$$\|(F(\mathbf{z}_1, \mathbf{x}_0) + \mathbf{z}_1) - (F(\mathbf{z}_2, \mathbf{x}_0) + \mathbf{z}_2)\| = \|g_z(\mathbf{z}_1) - g_z(\mathbf{z}_2)\| < \|\mathbf{z}_1 - \mathbf{z}_2\|.$$

Therefore, $h_{\mathbf{x}_0}(\mathbf{z})$ has an unique fixed point, denoted by $f(\mathbf{x}_0)$ :

$$h_{\mathbf{x}_0}(f(\mathbf{x}_0)) = f(\mathbf{x}_0) \Leftrightarrow F(f(\mathbf{x}_0), \mathbf{x}_0) = 0$$

Similarly, we also have: $\forall \mathbf{z}_0 \in \mathbb{R}^d$, there exists an unique $g(\mathbf{z}_0)$ satisfying $F(\mathbf{z}_0, g(\mathbf{z}_0)) = 0$.

Moreover, Let $\mathbf{z}_0 = f(\mathbf{x}_0)$, we have $F(f(\mathbf{x}_0), g(f(\mathbf{x}_0))) = 0$. By the uniqueness, we have $g(f(\mathbf{x}_0)) = \mathbf{x}_0, \forall \mathbf{x}_0 \in \mathbb{R}^d$. Similarly, $f(g(\mathbf{x}_0)) = \mathbf{x}_0, \forall \mathbf{x}_0 \in \mathbb{R}^d$. Therefore, $f$ is unique and invertible. □

A.2   PROOF FOR THEOREM 2

We denote $\mathcal{D}$ as the set of all bi-Lipschitz $C^1$-diffeomorphisms from $\mathbb{R}^d$ to $\mathbb{R}^d$.

Firstly, we prove Lemma 1 in the main text.

*Proof.* (**Lemma 1**). $\forall f \in \mathcal{R}$, we have

$$\sup_{\mathbf{x} \in \mathbb{R}^d} \|J_f(\mathbf{x}) - I\|_2^2 < 1,$$

which is equivalent to

$$\sup_{\mathbf{x} \in \mathbb{R}^d, \mathbf{v} \in \mathbb{R}^d, \|\mathbf{v}\|_2 = 1} \|(J_f(\mathbf{x}) - I)\mathbf{v}\|_2^2 < 1 \text{ (Definition of operator norm.)}$$

$$\sup_{\mathbf{x} \in \mathbb{R}^d, \mathbf{v} \in \mathbb{R}^d, \|\mathbf{v}\|_2 = 1} \mathbf{v}^T (J_f^T(\mathbf{x}) - I)(J_f(\mathbf{x}) - I)\mathbf{v} < 1$$

$$\sup_{\mathbf{x} \in \mathbb{R}^d, \mathbf{v} \in \mathbb{R}^d, \|\mathbf{v}\|_2 = 1} \mathbf{v}^T J_f^T(\mathbf{x}) J_f(\mathbf{x}) \mathbf{v} - 2\mathbf{v}^T J_f(\mathbf{x}) \mathbf{v} < 0$$

Note that $J_f(\mathbf{x})$ is nonsingular, so $\forall \mathbf{x}, \mathbf{v} \in \mathbb{R}^d, \|\mathbf{v}\|_2 = 1$, we have $\mathbf{v}^T J_f^T(\mathbf{x}) J_f(\mathbf{x}) \mathbf{v} > 0$. Thus,

$$0 > \sup_{\mathbf{x} \in \mathbb{R}^d, \mathbf{v} \in \mathbb{R}^d, \|\mathbf{v}\|_2 = 1} \mathbf{v}^T J_f^T(\mathbf{x}) J_f(\mathbf{x}) \mathbf{v} - 2\mathbf{v}^T J_f(\mathbf{x}) \mathbf{v} \geq \sup_{\mathbf{x} \in \mathbb{R}^d, \mathbf{v} \in \mathbb{R}^d, \|\mathbf{v}\|_2 = 1} -2\mathbf{v}^T J_f(\mathbf{x}) \mathbf{v}$$

So we have

$$\inf_{\mathbf{x} \in \mathbb{R}^d, \mathbf{v} \in \mathbb{R}^d, \|\mathbf{v}\|_2 = 1} \mathbf{v}^T J_f(\mathbf{x}) \mathbf{v} > 0.$$

Note that the converse is not true, because $\mathbf{v}^T J_f(\mathbf{x}) \mathbf{v} > 0$ does not restrict the upper bound of Lipschitz constant of $f$. For example, when $f(\mathbf{x}) = m\mathbf{x}$ where $m$ is a positive real number, we have

$$\inf_{\mathbf{x} \in \mathbb{R}^d, \mathbf{v} \in \mathbb{R}^d, \|\mathbf{v}\|_2 = 1} \mathbf{v}^T J_f(\mathbf{x}) \mathbf{v} = \inf_{\mathbf{x} \in \mathbb{R}^d, \mathbf{v} \in \mathbb{R}^d, \|\mathbf{v}\|_2 = 1} \mathbf{v}^T (mI)\mathbf{v} = m > 0$$

However, $m$ can be any large positive number. So we have $\mathcal{R} \subsetneq \mathcal{F}$.   $\square$

**Lemma 2.** $\forall f \in \mathcal{D}$, if

$$\inf_{\substack{\mathbf{x} \in \mathbb{R}^d, \mathbf{v} \in \mathbb{R}^d, \\ \|\mathbf{v}\|_2 = 1}} \mathbf{v}^T J_f(\mathbf{x}) \mathbf{v} > 0, \tag{17}$$

*then*

$$\inf_{\substack{\mathbf{x} \in \mathbb{R}^d, \mathbf{v} \in \mathbb{R}^d, \\ \|\mathbf{v}\|_2 = 1}} \mathbf{v}^T J_{f^{-1}}(\mathbf{x}) \mathbf{v} > 0, \tag{18}$$

*Proof.* (Proof of Lemma 2). By *Inverse Function Theorem*,

$$J_{f^{-1}}(\mathbf{x}) = J_f^{-1}(f^{-1}(\mathbf{x})).$$

Because $f$ is from $\mathbb{R}^d$ to $\mathbb{R}^d$, we have

$$\inf_{\mathbf{x} \in \mathbb{R}^d, \mathbf{v} \in \mathbb{R}^d, \|\mathbf{v}\|_2 = 1} \mathbf{v}^T J_{f^{-1}}(\mathbf{x}) \mathbf{v} = \inf_{\mathbf{x} \in \mathbb{R}^d, \mathbf{v} \in \mathbb{R}^d, \|\mathbf{v}\|_2 = 1} \mathbf{v}^T J_f^{-1}(f^{-1}(\mathbf{x})) \mathbf{v}$$

$$= \inf_{\mathbf{x} \in \mathbb{R}^d, \mathbf{v} \in \mathbb{R}^d, \|\mathbf{v}\|_2 = 1} \mathbf{v}^T J_f^{-1}(\mathbf{x}) \mathbf{v}$$

Let $\mathbf{u} = J_f^{-1}(\mathbf{x})\mathbf{v}$ and $\mathbf{v}_0 = \frac{\mathbf{u}}{\|\mathbf{u}\|_2}$, we have $\|\mathbf{v}_0\|_2 = 1$, and

$$\mathbf{v}^T J_f^{-1}(\mathbf{x}) \mathbf{v} = \mathbf{u}^T J_f^T(\mathbf{x}) \mathbf{u} = \mathbf{u}^T J_f(\mathbf{x}) \mathbf{u} = \|\mathbf{u}\|_2^2 \mathbf{v}_0^T J_f(\mathbf{x}) \mathbf{v}_0.$$

The above equation uses this fact: for a real $d \times d$ matrix $A$, $\forall \mathbf{x} \in \mathbb{R}^d, \mathbf{x}^T A \mathbf{x} = (\mathbf{x}^T A \mathbf{x})^T = \mathbf{x}^T A^T \mathbf{x}$ because $\mathbf{x}^T A \mathbf{x} \in \mathbb{R}$.

Note that $f$ is Lipschitz continuous, $\|J_f(\mathbf{x})\|_2 \leq \text{Lip}(f)$. So

$$1 = \|\mathbf{v}\|_2 \leq \|J_f(\mathbf{x})\|_2 \|\mathbf{u}\|_2 \leq \text{Lip}(f)\|\mathbf{u}\|_2,$$

which means

$$\|\mathbf{u}\|_2 \geq \frac{1}{\text{Lip}(f)}.$$

Thus,

$$
\begin{aligned}
\inf_{\mathbf{x}\in\mathbb{R}^d,\mathbf{v}\in\mathbb{R}^d,\|\mathbf{v}\|_2=1} \mathbf{v}^T J_f^{-1}(\mathbf{x})\mathbf{v} &= \inf_{\mathbf{x}\in\mathbb{R}^d,\mathbf{v}\in\mathbb{R}^d,\|\mathbf{v}\|_2=1} \|\mathbf{u}\|_2^2 \mathbf{v}_0^T J_f(\mathbf{x})\mathbf{v}_0 \\
&\geq \inf_{\mathbf{x}\in\mathbb{R}^d,\mathbf{u}\in\mathbb{R}^d,\|J_f(\mathbf{x})\mathbf{u}\|_2=1} \|\mathbf{u}\|_2^2 \inf_{\mathbf{x}\in\mathbb{R}^d,\mathbf{v}_0\in\mathbb{R}^d,\|\mathbf{v}_0\|_2=1} \mathbf{v}_0^T J_f(\mathbf{x})\mathbf{v}_0 \\
&\geq \frac{1}{\text{Lip}(f)^2} \inf_{\mathbf{x}\in\mathbb{R}^d,\mathbf{v}\in\mathbb{R}^d,\|\mathbf{v}\|_2=1} \mathbf{v}^T J_f(\mathbf{x})\mathbf{v} \\
&> 0
\end{aligned}
$$

$\square$

**Lemma 3.** $\forall f \in \mathcal{D}$, if $f^{-1} \in \mathcal{R}$, we have

$$\inf_{\substack{\mathbf{x}\in\mathbb{R}^d,\mathbf{v}\in\mathbb{R}^d, \\ \|\mathbf{v}\|_2=1}} \mathbf{v}^T J_f(\mathbf{x})\mathbf{v} > 0. \tag{19}$$

*Proof.* (Proof of Lemma 3). $\forall f \in \mathcal{D}$, if $f^{-1} \in \mathcal{R}$, then from Lemma 1, we have

$$\inf_{\mathbf{x}\in\mathbb{R}^d,\mathbf{v}\in\mathbb{R}^d,\|\mathbf{v}\|_2=1} \mathbf{v}^T J_{f^{-1}}(\mathbf{x})\mathbf{v} > 0.$$

Note that $f^{-1} \in \mathcal{D}$, from Lemma 2 we have

$$\inf_{\mathbf{x}\in\mathbb{R}^d,\mathbf{v}\in\mathbb{R}^d,\|\mathbf{v}\|_2=1} \mathbf{v}^T J_f(\mathbf{x})\mathbf{v} > 0.$$

$\square$

**Lemma 4.** $\forall f \in \mathcal{D}$, if

$$\inf_{\mathbf{x}\in\mathbb{R}^d,\mathbf{v}\in\mathbb{R}^d,\|\mathbf{v}\|_2=1} \mathbf{v}^T J_f(\mathbf{x})\mathbf{v} > 0,$$

*then $\exists\, \alpha_0 > 0$, s.t. $\forall\, 0 < \alpha < \alpha_0$,*

$$\sup_{\mathbf{x}\in\mathbb{R}^d} \|\alpha J_f(\mathbf{x}) - I\|_2 < 1.$$

*Proof.* (Proof of Lemma 4). Note that $f$ is Lipschitz continuous, so $\text{Lip}(f) = \sup_{\mathbf{x}\in\mathbb{R}^d} \|J_f(\mathbf{x})\|_2$. Denote

$$\beta = \inf_{\mathbf{x}\in\mathbb{R}^d,\mathbf{v}\in\mathbb{R}^d,\|\mathbf{v}\|_2=1} \mathbf{v}^T J_f(\mathbf{x})\mathbf{v}.$$

And let

$$\alpha_0 = \frac{\beta}{\text{Lip}(f)^2} > 0.$$

$\forall\, 0 < \alpha < \alpha_0$, we have

$$\sup_{\mathbf{x}\in\mathbb{R}^d} \|\alpha J_f(\mathbf{x}) - I\|_2^2 = \sup_{\mathbf{x}\in\mathbb{R}^d,\mathbf{v}\in\mathbb{R}^d,\|\mathbf{v}\|_2=1} \mathbf{v}^T(\alpha J_f^T(\mathbf{x}) - I)(\alpha J_f(\mathbf{x}) - I)\mathbf{v}$$

$$= 1 + \sup_{\mathbf{x}\in\mathbb{R}^d,\mathbf{v}\in\mathbb{R}^d,\|\mathbf{v}\|_2=1} \alpha^2\mathbf{v}^T J_f^T(\mathbf{x})J_f(\mathbf{x})\mathbf{v} - 2\alpha\mathbf{v}^T J_f(\mathbf{x})\mathbf{v}$$

$$\leq 1 + \alpha^2 \sup_{\mathbf{x}\in\mathbb{R}^d,\mathbf{v}\in\mathbb{R}^d,\|\mathbf{v}\|_2=1} \mathbf{v}^T J_f^T(\mathbf{x})J_f(\mathbf{x})\mathbf{v}$$

$$+ 2\alpha \sup_{\mathbf{x}\in\mathbb{R}^d,\mathbf{v}\in\mathbb{R}^d,\|\mathbf{v}\|_2=1} \left(-\mathbf{v}^T J_f(\mathbf{x})\mathbf{v}\right)$$

$$= 1 + \alpha^2 \sup_{\mathbf{x}\in\mathbb{R}^d,\mathbf{v}\in\mathbb{R}^d,\|\mathbf{v}\|_2=1} \mathbf{v}^T J_f^T(\mathbf{x})J_f(\mathbf{x})\mathbf{v}$$

$$- 2\alpha \inf_{\mathbf{x}\in\mathbb{R}^d,\mathbf{v}\in\mathbb{R}^d,\|\mathbf{v}\|_2=1} \mathbf{v}^T J_f(\mathbf{x})\mathbf{v}$$

$$= 1 + \alpha^2 \sup_{\mathbf{x}\in\mathbb{R}^d} \|J_f(\mathbf{x})\|_2^2 - 2\alpha\beta$$

$$= 1 + \alpha(\alpha\mathrm{Lip}(f)^2 - 2\beta)$$

$$< 1 + \alpha(\alpha_0\mathrm{Lip}(f)^2 - 2\beta)$$

$$= 1 - \alpha\beta$$

$$< 1.$$

The above equation uses this fact: for a real $d\times d$ matrix $A$, $\forall \mathbf{x}\in\mathbb{R}^d$, $\mathbf{x}^T A\mathbf{x} = (\mathbf{x}^T A\mathbf{x})^T = \mathbf{x}^T A^T\mathbf{x}$ because $\mathbf{x}^T A\mathbf{x}\in\mathbb{R}$. □

*Proof.* (**Theorem 2**) Denote

$$\mathcal{P} = \{f \in \mathcal{D} \mid \exists f_1, f_2 \in \mathcal{D}, f = f_2 \circ f_1, \text{where}$$

$$\inf_{\mathbf{x}\in\mathbb{R}^d,\mathbf{v}\in\mathbb{R}^d,\|\mathbf{v}\|_2=1} \mathbf{v}^T J_{f_1}(\mathbf{x})\mathbf{v} > 0, \inf_{\mathbf{x}\in\mathbb{R}^d,\mathbf{v}\in\mathbb{R}^d,\|\mathbf{v}\|_2=1} \mathbf{v}^T J_{f_2}(\mathbf{x})\mathbf{v} > 0\}.$$

Firstly, we show that $\mathcal{I} \subset \mathcal{P}$. $\forall f \in \mathcal{I}$, assume $f = f_2 \circ f_1$, where $f_1 \in \mathcal{R}$ and $f_2^{-1} \in \mathcal{R}$. By Lemma 1 and Lemma 3, we have

$$\inf_{\mathbf{x}\in\mathbb{R}^d,\mathbf{v}\in\mathbb{R}^d,\|\mathbf{v}\|_2=1} \mathbf{v}^T J_{f_1}(\mathbf{x})\mathbf{v} > 0,$$

$$\inf_{\mathbf{x}\in\mathbb{R}^d,\mathbf{v}\in\mathbb{R}^d,\|\mathbf{v}\|_2=1} \mathbf{v}^T J_{f_2}(\mathbf{x})\mathbf{v} > 0.$$

Thus, $f \in \mathcal{P}$. So $\mathcal{I} \subset \mathcal{P}$.

Next, we show that $\mathcal{P} \subset \mathcal{I}$. $\forall f \in \mathcal{P}$, assume $f = f_2 \circ f_1$, where

$$\inf_{\mathbf{x}\in\mathbb{R}^d,\mathbf{v}\in\mathbb{R}^d,\|\mathbf{v}\|_2=1} \mathbf{v}^T J_{f_1}(\mathbf{x})\mathbf{v} > 0,$$

$$\inf_{\mathbf{x}\in\mathbb{R}^d,\mathbf{v}\in\mathbb{R}^d,\|\mathbf{v}\|_2=1} \mathbf{v}^T J_{f_2}(\mathbf{x})\mathbf{v} > 0.$$

From Lemma 2, we have

$$\inf_{\mathbf{x}\in\mathbb{R}^d,\mathbf{v}\in\mathbb{R}^d,\|\mathbf{v}\|_2=1} \mathbf{v}^T J_{f_2^{-1}}(\mathbf{x})\mathbf{v} > 0.$$

From Lemma 4, $\exists\, \alpha_1 > 0, \alpha_2 > 0$, s.t. $\forall\, 0 < \alpha < \min\{\alpha_1, \alpha_2\}$,

$$\sup_{\mathbf{x}\in\mathbb{R}^d} \|\alpha J_{f_1}(\mathbf{x}) - I\|_2 < 1,$$

$$\sup_{\mathbf{x}\in\mathbb{R}^d} \|\alpha J_{f_2^{-1}}(\mathbf{x}) - I\|_2 < 1.$$

Let $\alpha = \frac{1}{2}\min\{\alpha_1, \alpha_2\}$. Let $g = g_2 \circ g_1$, where

$$g_1(\mathbf{x}) = \alpha f_1(\mathbf{x}),$$

$$g_2(\mathbf{x}) = f_2(\frac{\mathbf{x}}{\alpha}).$$

We have $g(\mathbf{x}) = f_2(\frac{\alpha f_1(\mathbf{x})}{\alpha}) = f(\mathbf{x})$, and

$$J_{g_1}(\mathbf{x}) = \alpha J_{f_1}(\mathbf{x}),$$
$$g_2^{-1}(\mathbf{x}) = \alpha f_2^{-1}(\mathbf{x}),$$
$$J_{g_2^{-1}}(\mathbf{x}) = \alpha J_{f_2^{-1}}(\mathbf{x}).$$

So we have

$$\sup_{\mathbf{x} \in \mathbb{R}^d} \|J_{g_1}(\mathbf{x}) - I\|_2 = \sup_{\mathbf{x} \in \mathbb{R}^d} \|\alpha J_{f_1}(\mathbf{x}) - I\|_2 < 1,$$
$$\sup_{\mathbf{x} \in \mathbb{R}^d} \|J_{g_2^{-1}}(\mathbf{x}) - I\|_2 = \sup_{\mathbf{x} \in \mathbb{R}^d} \|\alpha J_{f_2^{-1}}(\mathbf{x}) - I\|_2 < 1.$$

Thus, $g_1 \in \mathcal{R}$ and $g_2^{-1} \in \mathcal{R}$ and $f = g_2 \circ g_1$. So $f \in \mathcal{I}$. Therefore, $\mathcal{P} \subset \mathcal{I}$.

In conclusion, $\mathcal{I} = \mathcal{P}$. $\qquad \square$

### A.3 Proof for Theorem 3

Firstly, we prove a lemma of bi-Lipschitz continuous functions.

**Lemma 5.** *If $f : (\mathbb{R}^d, \|\cdot\|) \to (\mathbb{R}^d, \|\cdot\|)$ is bi-Lipschitz continuous, then*

$$\frac{1}{\mathrm{Lip}(f^{-1})} \leq \frac{\|f(\mathbf{x}_1) - f(\mathbf{x}_2)\|}{\|\mathbf{x}_1 - \mathbf{x}_2\|} \leq \mathrm{Lip}(f), \ \forall \mathbf{x}_1, \mathbf{x}_2 \in \mathbb{R}^d, \mathbf{x}_1 \neq \mathbf{x}_2.$$

*Proof.* (Proof of Lemma 5). $\forall \mathbf{x}_1, \mathbf{x}_2 \in \mathbb{R}^d, \mathbf{x}_1 \neq \mathbf{x}_2$, we have

$$\|f(\mathbf{x}_1) - f(\mathbf{x}_2)\| \leq \mathrm{Lip}(f)\|\mathbf{x}_1 - \mathbf{x}_2\|$$
$$\|\mathbf{x}_1 - \mathbf{x}_2\| = \|f^{-1}(f(\mathbf{x}_1)) - f^{-1}(f(\mathbf{x}_2))\| \leq \mathrm{Lip}(f^{-1})\|f(\mathbf{x}_1) - f(\mathbf{x}_2)\|$$

Thus, we get the results. $\qquad \square$

Assume a residual flow $f = f_L \circ \cdots \circ f_1$ where each layer $f_l$ is an invertible residual network:

$$f_l(\mathbf{x}) = \mathbf{x} + g_l(\mathbf{x}), \ \mathrm{Lip}(g_l) \leq \kappa < 1.$$

Thus, each layer $f_l$ is bi-Lipschitz and it follows by Behrmann et al. (2019) and Lemma 5 that

$$1 - \kappa \leq \frac{\|f_l(\mathbf{x}_1) - f_l(\mathbf{x}_2)\|}{\|\mathbf{x}_1 - \mathbf{x}_2\|} \leq 1 + \kappa < 2^L, \ \forall \mathbf{x}_1, \mathbf{x}_2 \in \mathbb{R}^d, \mathbf{x}_1 \neq \mathbf{x}_2. \tag{20}$$

By multiplying all the inequalities, we can get a bound of the bi-Lipschitz property for ResFlows, as shown in Lemma 6.

**Lemma 6.** *For ResFlows built by $f = f_L \circ \cdots \circ f_1$, where $f_l(\mathbf{x}) = \mathbf{x} + g_l(\mathbf{x}), \mathrm{Lip}(g_l) \leq \kappa < 1$, then*

$$(1 - \kappa)^L \leq \frac{\|f(\mathbf{x}_1) - f(\mathbf{x}_2)\|}{\|\mathbf{x}_1 - \mathbf{x}_2\|} \leq (1 + \kappa)^L, \ \forall \mathbf{x}_1, \mathbf{x}_2 \in \mathbb{R}^d, \mathbf{x}_1 \neq \mathbf{x}_2.$$

Next, we prove Theorem 3.

*Proof.* (**Theorem 3**) According to the definition of $\mathcal{P}(L, r)$, we have $\mathcal{P}(L, r) \subset \mathcal{F} \subset \mathcal{I}$.

$\forall \, 0 < \ell < \log_2(L)$, we have $L - 2^\ell > 0$. $\forall \, g \in \mathcal{R}_\ell$, by Lemma 6, we have

$$\|g(\mathbf{x}) - g(\mathbf{y})\|_2 \leq 2^\ell \|\mathbf{x} - \mathbf{y}\|_2, \forall \mathbf{x}, \mathbf{y} \in \mathcal{B}_r.$$

Thus, $\forall \, \mathbf{x}_0 \in \mathcal{B}_r$, we have

$$\begin{aligned}
\|f(\mathbf{x}) - g(\mathbf{x})\|_2 &= \|f(\mathbf{x}) - f(\mathbf{x}_0) + g(\mathbf{x}_0) - g(\mathbf{x}) + f(\mathbf{x}_0) - g(\mathbf{x}_0)\|_2 \\
&\geq \|f(\mathbf{x}) - f(\mathbf{x}_0)\|_2 - \|g(\mathbf{x}_0) - g(\mathbf{x}) + f(\mathbf{x}_0) - g(\mathbf{x}_0)\|_2 \\
&\geq \|f(\mathbf{x}) - f(\mathbf{x}_0)\|_2 - \|g(\mathbf{x}_0) - g(\mathbf{x})\|_2 - \|f(\mathbf{x}_0) - g(\mathbf{x}_0)\|_2 \\
&\geq (L - 2^\ell)\|\mathbf{x} - \mathbf{x}_0\|_2 - \|f(\mathbf{x}_0) - g(\mathbf{x}_0)\|_2
\end{aligned}$$

So

$$\sup_{\mathbf{x}\in\mathcal{B}_r} \|f(\mathbf{x}) - g(\mathbf{x})\|_2 \geq \sup_{\mathbf{x}\in\mathcal{B}_r} (L - 2^\ell)\|\mathbf{x} - \mathbf{x}_0\|_2 - \|f(\mathbf{x}_0) - g(\mathbf{x}_0)\|_2$$
$$\geq (L - 2^\ell)r - \|f(\mathbf{x}_0) - g(\mathbf{x}_0)\|_2$$

Notice that the inequality above is true for any $\mathbf{x}_0 \in \mathcal{B}_r$, so we have

$$\sup_{\mathbf{x}\in\mathcal{B}_r} \|f(\mathbf{x}) - g(\mathbf{x})\|_2 \geq \sup_{\mathbf{x}_0\in\mathcal{B}_r} (L - 2^\ell)r - \|f(\mathbf{x}_0) - g(\mathbf{x}_0)\|_2$$
$$= (L - 2^\ell)r - \inf_{\mathbf{x}_0\in\mathcal{B}_r} \|f(\mathbf{x}_0) - g(\mathbf{x}_0)\|_2$$
$$\geq (L - 2^\ell)r - \sup_{\mathbf{x}_0\in\mathcal{B}_r} \|f(\mathbf{x}_0) - g(\mathbf{x}_0)\|_2$$

Therefore,

$$\sup_{\mathbf{x}\in\mathcal{B}_r} \|f(\mathbf{x}) - g(\mathbf{x})\|_2 \geq \frac{r}{2}(L - 2^\ell), \forall g \in \mathcal{R}^\ell$$

So we get

$$\inf_{g\in\mathcal{R}^\ell} \sup_{\mathbf{x}\in\mathcal{B}_r} \|f(\mathbf{x}) - g(\mathbf{x})\|_2 \geq \frac{r}{2}(L - 2^\ell)$$

Because $\forall f \in \mathcal{P}(L, r)$, $\inf_{g\in\mathcal{R}_\ell} \sup_{\mathbf{x}\in\mathcal{B}_r} \|f(\mathbf{x}) - g(\mathbf{x})\|_2 > 0$, we have $\mathcal{R}^\ell \cap \mathcal{P}(L, r) = \varnothing$. □

### A.4 Proof for Equation 13

*Proof.* (**Equation 13**) By Change of Variable formula:

$$\log p(\mathbf{x}) = \log p(\mathbf{z}) + \log |\partial\mathbf{z}/\partial\mathbf{x}|,$$

Since $\mathbf{z} = f(\mathbf{x})$ is defined by the equation

$$F(\mathbf{z}, \mathbf{x}) = g_{\mathbf{x}}(\mathbf{x}) - g_{\mathbf{z}}(\mathbf{z}) + \mathbf{x} - \mathbf{z} = 0,$$

by Implicit function theorem, we have

$$\partial\mathbf{z}/\partial\mathbf{x} = J_f(\mathbf{x}) = -[J_{F,\mathbf{z}}(\mathbf{z})]^{-1}[J_{F,\mathbf{x}}(\mathbf{x})] = (I + J_{g_{\mathbf{z}}}(\mathbf{z}))^{-1}(I + J_{g_{\mathbf{x}}}(\mathbf{x})).$$

Thus,

$$\log |\partial\mathbf{z}/\partial\mathbf{x}| = \ln |\det(I + J_{g_{\mathbf{x}}}(\mathbf{x}))| - \ln |\det(I + J_{g_{\mathbf{z}}}(\mathbf{z}))|$$

Note that any eigenvalue $\lambda$ of $J_{g_{\mathbf{x}}}(\mathbf{x})$ satisfies $|\lambda| < \sigma(J_{g_{\mathbf{x}}}(\mathbf{x})) = \|J_{g_{\mathbf{x}}}(\mathbf{x})\|_2 < 1$, so $\lambda \in (-1, 1)$. Thus, $\det(I + J_{g_{\mathbf{x}}}(\mathbf{x})) > 0$. Similarly, $\det(I + J_{g_{\mathbf{z}}}(\mathbf{z})) > 0$. Therefore,

$$\log |\partial\mathbf{z}/\partial\mathbf{x}| = \ln \det(I + J_{g_{\mathbf{x}}}(\mathbf{x})) - \ln \det(I + J_{g_{\mathbf{z}}}(\mathbf{z}))$$

□

### A.5 Proof for Equation 16

*Proof.* (**Equation 16**) By implicitly differentiating two sides of $F(\mathbf{z}, \mathbf{x}; \theta) = 0$ by $\mathbf{x}$, we have

$$\frac{\partial g_{\mathbf{x}}(\mathbf{x}; \theta)}{\partial\mathbf{x}} - \frac{\partial g_{\mathbf{z}}(\mathbf{z}; \theta)}{\partial\mathbf{z}} \frac{\partial\mathbf{z}}{\partial\mathbf{x}} + I - \frac{\partial\mathbf{z}}{\partial\mathbf{x}} = 0,$$

So we have

$$\frac{\partial\mathbf{z}}{\partial\mathbf{x}} = \left(I + \frac{\partial g_{\mathbf{z}}(\mathbf{z}; \theta)}{\partial\mathbf{z}}\right)^{-1} \left(I + \frac{\partial g_{\mathbf{x}}(\mathbf{x}; \theta)}{\partial\mathbf{x}}\right)$$
$$= J_G^{-1}(\mathbf{z}) \frac{\partial F(\mathbf{z}, \mathbf{x}; \theta)}{\partial\mathbf{x}}$$

By implicitly differentiating two sides of $F(\mathbf{z}, \mathbf{x}; \theta) = 0$ by $\theta$, we have

$$\frac{\partial g_{\mathbf{x}}(\mathbf{x}; \theta)}{\partial \theta} - \frac{\partial g_{\mathbf{z}}(\mathbf{z}; \theta)}{\partial \theta} - \frac{\partial g_{\mathbf{z}}(\mathbf{z}; \theta)}{\partial \mathbf{z}} \frac{\partial \mathbf{z}}{\partial \theta} - \frac{\partial \mathbf{z}}{\partial \theta} = 0,$$

So we have

$$\frac{\partial \mathbf{z}}{\partial \theta} = \left( I + \frac{\partial g_{\mathbf{z}}(\mathbf{z}; \theta)}{\partial \mathbf{z}} \right)^{-1} \left( \frac{\partial g_{\mathbf{x}}(\mathbf{x}; \theta)}{\partial \theta} - \frac{\partial g_{\mathbf{z}}(\mathbf{z}; \theta)}{\partial \theta} \right)$$

$$= J_G^{-1}(\mathbf{z}) \frac{\partial F(\mathbf{z}, \mathbf{x}; \theta)}{\partial \theta}$$

Therefore, the gradient from $\mathbf{z}$ to $(\cdot)$ is

$$\frac{\partial \mathcal{L}}{\partial \mathbf{z}} \frac{\partial \mathbf{z}}{\partial (\cdot)} = \frac{\partial \mathcal{L}}{\partial \mathbf{z}} J_G^{-1}(\mathbf{z}) \frac{\partial F(\mathbf{z}, \mathbf{x}; \theta)}{\partial (\cdot)}.$$

$\square$

## B  OTHER PROPERTIES OF IMPLICIT FLOWS

In this section, we propose some other properties of ImpFlows.

**Lemma 7.** *For a single implicit flow $f \in \mathcal{I}$, assume that $f = f_2^{-1} \circ f_1$, where*

$$f_1(\mathbf{x}) = \mathbf{x} + g_1(\mathbf{x}), \ \text{Lip}(g_1) \le \kappa < 1, \tag{21}$$
$$f_2(\mathbf{x}) = \mathbf{x} + g_2(\mathbf{x}), \ \text{Lip}(g_2) \le \kappa < 1, \tag{22}$$

*then*

$$\frac{1 - \kappa}{1 + \kappa} \le \frac{\|f(\mathbf{x}_1) - f(\mathbf{x}_2)\|}{\|\mathbf{x}_1 - \mathbf{x}_2\|} \le \frac{1 + \kappa}{1 - \kappa}, \ \forall \mathbf{x}_1, \mathbf{x}_2 \in \mathbb{R}^d, \mathbf{x}_1 \ne \mathbf{x}_2. \tag{23}$$

*Proof.* (Proof of **Lemma 7**) According to Eqn. (20), we have

$$1 - \kappa \le \frac{\|f_1(\mathbf{x}_1) - f_1(\mathbf{x}_2)\|}{\|\mathbf{x}_1 - \mathbf{x}_2\|} \le 1 + \kappa, \ \forall \mathbf{x}_1, \mathbf{x}_2 \in \mathbb{R}^d, \mathbf{x}_1 \ne \mathbf{x}_2, \tag{24}$$

$$\frac{1}{1 + \kappa} \le \frac{\|f_2^{-1}(\mathbf{x}_1) - f_2^{-1}(\mathbf{x}_2)\|}{\|\mathbf{x}_1 - \mathbf{x}_2\|} \le \frac{1}{1 - \kappa}, \ \forall \mathbf{x}_1, \mathbf{x}_2 \in \mathbb{R}^d, \mathbf{x}_1 \ne \mathbf{x}_2. \tag{25}$$

By multiplying these two inequalities, we can get the results. $\square$

**Theorem 4.** *(Limitation of the single ImpFlow).*

$$\mathcal{I} \subset \{f : f \in \mathcal{D}, \forall \mathbf{x} \in \mathbb{R}^d, \lambda(J_f(\mathbf{x})) \cap \mathbb{R}^- = \varnothing\}, \tag{26}$$

*where $\lambda(A)$ denotes the set of all eigenvalues of matrix $A$.*

*Proof.* (Proof of **Theorem 4**)

Proof by contradiction. Assume $\exists f \in \mathcal{I}$ and $\mathbf{x} \in \mathbb{R}^d$, s.t. $\exists \lambda \in \lambda(J_f(\mathbf{x})), \lambda < 0$.

There exists a vector $\mathbf{u} \ne 0, J_f(\mathbf{x})\mathbf{u} = \lambda\mathbf{u}$. By Theorem 2, $\exists f_1, f_2 \in \mathcal{F}, f = f_2 \circ f_1$, hence $J_f(\mathbf{x}) = J_{f_2}(f_1(\mathbf{x}))J_{f_1}(\mathbf{x})$. We denote $A := J_{f_2}(f_1(\mathbf{x})), B := J_{f_1}(\mathbf{x})$. Since $f_1, f_2 \in \mathcal{F}$, we have

$$\mathbf{v}^T A \mathbf{v} > 0, \mathbf{w}^T B \mathbf{w} > 0, \forall \mathbf{v}, \mathbf{w} \ne 0, \mathbf{v}, \mathbf{w} \in \mathbb{R}^d.$$

Note that B is the Jacobian of a bi-Lipschitz function at a single point, so B is non-singular. As $\mathbf{u} \ne 0$, we have $B\mathbf{u} \ne 0$. Thus,

$$(B\mathbf{u})^T A(B\mathbf{u}) = (B\mathbf{u})^T((AB)\mathbf{u}) = \lambda\mathbf{u}^T B^T \mathbf{u} = \lambda\mathbf{u}^T B\mathbf{u}$$

The last equation uses this fact: for a real $d \times d$ matrix $A$, $\forall \mathbf{x} \in \mathbb{R}^d, \mathbf{x}^T A\mathbf{x} = (\mathbf{x}^T A\mathbf{x})^T = \mathbf{x}^T A^T \mathbf{x}$ because $\mathbf{x}^T A\mathbf{x} \in \mathbb{R}$. Note that the left side is positive, and the right side is negative. It's a contradiction. $\square$

Therefore, $\mathcal{I}$ cannot include all the bi-Lipschitz $C^1$-diffeomorphisms. As a corollary, we have $\mathcal{R}_3 \not\subset \mathcal{I}$.

**Corollary 2.** $\mathcal{R}_3 \not\subset \mathcal{I}$.

*Proof.* (Proof for **Corollary 2**) Consider three linear functions in $\mathcal{R}$:

$$f_1(\mathbf{x}) = \mathbf{x} + \begin{pmatrix} -0.46 & -0.20 \\ 0.85 & 0.00 \end{pmatrix} \mathbf{x}$$

$$f_2(\mathbf{x}) = \mathbf{x} + \begin{pmatrix} -0.20 & -0.70 \\ 0.30 & -0.60 \end{pmatrix} \mathbf{x}$$

$$f_3(\mathbf{x}) = \mathbf{x} + \begin{pmatrix} -0.50 & -0.60 \\ -0.20 & -0.55 \end{pmatrix} \mathbf{x}$$

We can get that $f = f_1 \circ f_2 \circ f_3$ is in $\mathcal{R}_3$, and $f$ is also a linear function with Jacobian $\begin{pmatrix} 0.2776 & -0.4293 \\ 0.5290 & -0.6757 \end{pmatrix}$ However, this is a matrix with two negative eigenvalues: -0.1881, -0.2100. Hence $f$ is not in $\mathcal{I}$. Therefore, $\mathcal{R}_3 \not\subset \mathcal{I}$. $\qquad\square$

## C  COMPUTATION

### C.1  APPROXIMATE INVERSE JACOBIAN

The exact computation for the Jacobian inverse term costs much for high dimension tasks. We use the similar technique in Bai et al. (2019) to compute $\frac{\partial \mathcal{L}}{\partial \mathbf{z}} J_G^{-1}(\mathbf{z})$: solving the linear system of variable $\mathbf{y}$:

$$J_G^T(\mathbf{z})\mathbf{y}^T = (\frac{\partial \mathcal{L}}{\partial \mathbf{z}})^T, \tag{27}$$

where the left hand side is a vector-Jacobian product and it can be efficiently computed by autograd packages foy any $\mathbf{y}$ without computing the Jacobian matrix. In this work, we also use Broyden's method to solve the root, the same as methods in the forward pass, where the tolerance bound for the stop criterion is $\epsilon_b$.

**Remark.** Although the forward, inverse and backward pass of ImpFlows all need to solve the root of some equation, we can choose small enough $\epsilon_f$ and $\epsilon_b$ to ensure the approximation error is small enough. Thus, there is a trade-off between computation costs and approximation error. In practice, we use $\epsilon_f = 10^{-6}$ and $\epsilon_b = 10^{-10}$ and empirically does not observe any error accumulation. Note that such approximation is rather different from the variational inference technique in Chen et al. (2020); Nielsen et al. (2020), because we only focus on the exact log density itself.

### C.2  COMPUTATION TIME

We evaluate the average computation time for the model trained on CIFAR10 in Table 3 on a single Tesla P100 (SXM2-16GB). See Table 4 for the details. For a fair comparision, the forward (inference) time in the training phase of ImpFlow is comparable to that of ResFlow because the log-determinant term is the main cost. The backward time of ImpFlow costs more than that of ResFlow because it requires to rewrite the backward method in PyTorch to solve the linear equation. The training time includes forward, backward and other operations (such as the Lipschitz iterations for spectral normalization). We use the same method as the release code of ResFlows (fixed-point iterations with tolerance $10^{-5}$) for the sample phase. The sample time of ImpFlow is less than that of ResFlow because the inverse of $L$-block ImpFlow needs to solve $L$ fixed points while the inverse of $2L$-block ResFlow needs to solve $2L$ fixed points. Fast sampling is particularly desirable since it is the main advantage of flow-based models over autoregressive models.

Also, we evaluate the average Broyden's method iterations and the average function evaluation times during the Broyden's method. See Table 5 for the details.

Table 4: Single-batch computation time (seconds) for ResFlow and ImpFlow in Table 3 on a single Tesla P100 (SXM2-16GB).

| $c$ | Model | Forward (Inference) | | | Backward | | Training | Sample |
|---|---|---|---|---|---|---|---|---|
| 0.5 | ImpFlow | Fixed-point | Log-det | Others | Inv-Jacob | Others | 4.152 | 0.138 |
| | | 0.445 | 2.370 | 0.090 | 0.562 | 0.441 | | |
| | | | 2.905 | | 1.003 | | | |
| | ResFlow | | 2.656 | | 0.031 | | 2.910 | 0.229 |
| 0.6 | ImpFlow | Fixed-point | Log-det | Others | Inv-Jacob | Others | 4.415 | 0.159 |
| | | 0.497 | 2.356 | 0.120 | 0.451 | 0.800 | | |
| | | | 2.973 | | 1.251 | | | |
| | ResFlow | | 2.649 | | 0.033 | | 2.908 | 0.253 |
| 0.7 | ImpFlow | Fixed-point | Log-det | Others | Inv-Jacob | Others | 4.644 | 0.181 |
| | | 0.533 | 2.351 | 0.157 | 0.525 | 0.887 | | |
| | | | 3.041 | | 1.412 | | | |
| | ResFlow | | 2.650 | | 0.030 | | 2.908 | 0.312 |
| 0.8 | ImpFlow | Fixed-point | Log-det | Others | Inv-Jacob | Others | 4.881 | 0.206 |
| | | 0.602 | 2.364 | 0.139 | 0.641 | 0.943 | | |
| | | | 3.105 | | 1.584 | | | |
| | ResFlow | | 2.655 | | 0.030 | | 2.910 | 0.374 |
| 0.9 | ImpFlow | Fixed-point | Log-det | Others | Inv-Jacob | Others | 5.197 | 0.258 |
| | | 0.707 | 2.357 | 0.137 | 0.774 | 1.033 | | |
| | | | 3.201 | | 1.807 | | | |
| | ResFlow | | 2.653 | | 0.030 | | 2.916 | 0.458 |

Table 5: Single-batch iterations of Broyden's method during forward and backward pass for a single block of ImpFlow in Table 3.

| $c$ | | Broyden's Method Iterations | Function Evaluations |
|---|---|---|---|
| 0.5 | Forward | 7.2 | 8.2 |
| | Backward | 12.5 | 13.5 |
| 0.6 | Forward | 8.3 | 9.3 |
| | Backward | 14.9 | 15.9 |
| 0.7 | Forward | 9.4 | 10.4 |
| | Backward | 17.9 | 18.9 |
| 0.8 | Forward | 10.8 | 11.8 |
| | Backward | 22.4 | 23.4 |
| 0.9 | Forward | 12.9 | 13.9 |
| | Backward | 27.4 | 28.4 |

## C.3 NUMERICAL SENSITIVITY

We train a 20-block ImpFlow on POWER dataset with $\epsilon_f = 10^{-6}$ (see Appendix. D for detailed settings), and then test this model with different $\epsilon_f$ to see the numerical sensitivity of the fixed-point iterations. Table 6 shows that our model is not sensitive to $\epsilon_f$ in a fair range.

## C.4 TRAINING ALGORITHM

We state the training algorithms for both forward and backward processes in Algorithm 1 and Algorithm 2

---

**Algorithm 1:** Forward Algorithm For a Single-Block ImpFlow

---

**Require:** $g_{\mathbf{x};\theta}, g_{\mathbf{z};\theta}$ in Eqn. (3), stop criterion $\epsilon_f$.

**Input:** $\mathbf{x}$.

**Output:** $\mathbf{z} = f(\mathbf{x})$ and $\ln p(\mathbf{x})$, where $f$ is the implicit function defined by $g_{\mathbf{x};\theta}$ and $g_{\mathbf{z};\theta}$.

Define $h(\mathbf{z}) = F(\mathbf{z}, \mathbf{x}; \theta)$

$\mathbf{z} \leftarrow \mathbf{0}$

**while** $\|h(\mathbf{z})\|_2 \geq \epsilon_f$ **do**

    $B \leftarrow$ The estimated inverse Jacobian of $h(\mathbf{z})$ (e.g. by Broyden's method)

    $\alpha \leftarrow \text{LineSearch}(\mathbf{z}, h, B)$

    $\mathbf{z} \leftarrow \mathbf{z} - \alpha B h(\mathbf{z})$

**if** *training* **then**

    Esitamate $\ln \det(I + J_{g_{\mathbf{x}}}(\mathbf{x}; \theta))$ by Eqn. (15)

    Esitamate $\ln \det(I + J_{g_{\mathbf{z}}}(\mathbf{z}; \theta))$ by Eqn. (15)

**else**

    Esitamate $\ln \det(I + J_{g_{\mathbf{x}}}(\mathbf{x}; \theta))$ by Eqn. (14)

    Esitamate $\ln \det(I + J_{g_{\mathbf{z}}}(\mathbf{z}; \theta))$ by Eqn. (14)

Compute $\ln p(\mathbf{x})$ by Eqn. (13)

---

**Algorithm 2:** Backward Algorithm For a Single-Block ImpFlow

---

**Require:** $g_{\mathbf{x};\theta}, g_{\mathbf{z};\theta}$ in Eqn. (3), stop criterion $\epsilon_b$.

**Input:** $\mathbf{x}, \mathbf{z}, \frac{\partial \mathcal{L}}{\partial \mathbf{z}}$.

**Output:** The gradient for $\mathbf{x}$ and $\theta$ from $\mathbf{z}$, i.e. $\frac{\partial \mathcal{L}}{\partial \mathbf{z}} \frac{\partial \mathbf{z}}{\partial \mathbf{x}}$ and $\frac{\partial \mathcal{L}}{\partial \mathbf{z}} \frac{\partial \mathbf{z}}{\partial \theta}$.

Define $G(\mathbf{z}; \theta) = g_{\mathbf{z}}(\mathbf{z}; \theta) + \mathbf{z}$ and $h(\mathbf{y}) = \mathbf{y} J_G(\mathbf{z}) - \frac{\partial \mathcal{L}}{\partial \mathbf{z}}$

$\mathbf{y} \leftarrow \mathbf{0}$

**while** $\|h(\mathbf{y})\|_2 \geq \epsilon_b$ **do**

    $B \leftarrow$ The estimated inverse Jacobian of $h(\mathbf{y})$ (e.g. by Broyden's method)

    $\alpha \leftarrow \text{LineSearch}(\mathbf{y}, h, B)$

    $\mathbf{y} \leftarrow \mathbf{y} - \alpha B h(\mathbf{y})$

Compute $\frac{\partial F(\mathbf{z},\mathbf{x};\theta)}{\partial \mathbf{x}}$ and $\frac{\partial F(\mathbf{z},\mathbf{x};\theta)}{\partial \theta}$ by autograd packages.

$\frac{\partial \mathcal{L}}{\partial \mathbf{z}} \frac{\partial \mathbf{z}}{\partial \mathbf{x}} \leftarrow \mathbf{y} \frac{\partial F(\mathbf{z},\mathbf{x};\theta)}{\partial \mathbf{x}}$

$\frac{\partial \mathcal{L}}{\partial \mathbf{z}} \frac{\partial \mathbf{z}}{\partial \theta} \leftarrow \mathbf{y} \frac{\partial F(\mathbf{z},\mathbf{x};\theta)}{\partial \theta}$

---

Table 6: Average test log-likelihood (in nats) for different $\epsilon_f$ of ImpFlow on POWER dataset.

| $\epsilon_f$ | $10^{-8}$ | $10^{-7}$ | $10^{-6}$ | $10^{-5}$ | $10^{-4}$ | $10^{-3}$ | $10^{-2}$ | $10^{-1}$ |
|---|---|---|---|---|---|---|---|---|
| log-likelihood | 0.606 | 0.603 | 0.607 | 0.611 | 0.607 | 0.607 | 0.602 | 0.596 |

# D   NETWORK STRUCTURES

## D.1   1-D EXAMPLE

We specify the function (data) to be fitted is

$$f(x) = \begin{cases} 0.1x, & x < 0 \\ 10x, & x \geq 0 \end{cases}$$

For $\mathcal{I}$, we can construct a fully-connected neural network with ReLU activation and 3 parameters as following:

$$g_x(x) = \mathrm{ReLU}(-0.9x)$$
$$g_z(z) = -\sqrt{0.9}\mathrm{ReLU}(\sqrt{0.9}z)$$

The two networks can be implemented by spectral normalization. Assume the implicit function defined by Eqn. (3) using the above $g_x(x)$ and $g_z(z)$ is $f_{\mathcal{I}}$. Next we show that $f = f_{\mathcal{I}}$.

Let $f_1(x) = x + \mathrm{ReLU}(-0.9x)$ and $f_2(x) = x - \sqrt{0.9}\mathrm{ReLU}(\sqrt{0.9}x)$, we have $f_2^{-1}(x) = x + \mathrm{ReLU}(9x)$. Therefore, $f_{\mathcal{I}} = f_2^{-1} \circ f_1 = f$.

For every residual block of $\mathcal{R}, \mathcal{R}_2$ and $\mathcal{R}_3$, we train a 4-layer MLP with ReLU activation and 128 hidden units, and the Lipschitz coefficient for the spectral normalization is 0.99, and the iteration number for the spectral computation is 200. The objective function is

$$\min_\theta \mathbb{E}_{x\sim\mathrm{Unif}(-1,1)} \left[ (f_\theta(x) - f(x))^2 \right],$$

where $f_\theta$ is the function of 1 or 2 or 3 residual blocks. We use a batch size of 5000. We use the Adam optimizer, with learning rate $10^{-3}$ and weight decay $10^{-5}$. We train the model until convergence, on a single NVIDIA GeForce GTX 1080Ti.

The losses for $\mathcal{R}, \mathcal{R}_2$ and $\mathcal{R}_3$ are $5.25, 2.47, 0.32$, respectively.

## D.2   CLASSIFICATION

For the classification tasks, we remove all the BatchNorm layers which are inside of a certain Res-Block, and only maintain the BatchNorm layer in the downsampling layer. Moreover, as a single ImpFlow consists of two residual blocks with the same dimension of input and output, we replace the downsampling shortcut by a identity shortcut in each scale of ResNet-18, and add a downsampling layer (a convolutional layer) with BatchNorm after the two residual blocks of each scale. Thus, each scale consists of two ResBlocks with the same dimension of input and output, which (6.5M parameters) is different from the vanilla ResNet-18 architecture (11.2M parameters). Note that the "vanilla ResNet-18" in our main text is refered to the 6.5M-parameter architecture, which is the same as the versions for ResFlow and ImpFlow.

We use the comman settings: batch size of 128, Adam optimizer with learning rate $10^{-3}$ and no weight decay, and total epoch of 150. For the spectral normalization iterations, we use a error bound of $10^{-3}$, the same as Chen et al. (2019). We train every experiment on a single NVIDIA GeForce GTX 2080Ti.

## D.3   DENSITY MODELING ON TOY 2D DATA

Following the same settings as Chen et al. (2019), we use 4-layer multilayer perceptrons (MLP) with fully-connected layers of 128 hidden units. We use the Adam optimizer with learning rate of $10^{-3}$ and weight decay of $10^{-5}$. Moreover, we find that $\frac{1}{2\pi}\sin(2\pi\mathbf{x})$ is a better activation for this

task while maintain the property of 1-Lipschitz constant, so we use this activation function for all experiments, which can lead to faster convergence and better log-likelihood for both ResFlows and ImpFlows, as shown in Fig. 4.

We do not use any ActNorm or BatchNorm layers. For the log-determinant term, we use brute-force computation as in Chen et al. (2019). For the forward and backward, we use the Broyden's method to compute the roots, with $\epsilon_f = 10^{-6}$. The Lipschitz coefficient for spectral normalization is $0.999$, and the iteration number for spectral normalization is 20. The batch size is 5000, and we train 50000 iterations. The test batch size is 10000.

Also, we vary the network depth to see the difference between ImpFlow and ResFlow. For every depth $L$, we use an $L$-block ImpFlow and a $2L$-block ResFlow with the same settings as stated above, and train 3 times with different random seeds. As shown in Figure 5, the gap between ImpFlow and ResFlow shrinks as the depth grows deep, because the Lipschitz constant of ResFlow grows exponentially. Note that the dashed line is a 200-block ResFlow in Chen et al. (2019), and we tune our model better so that our models perform better with lower depth.

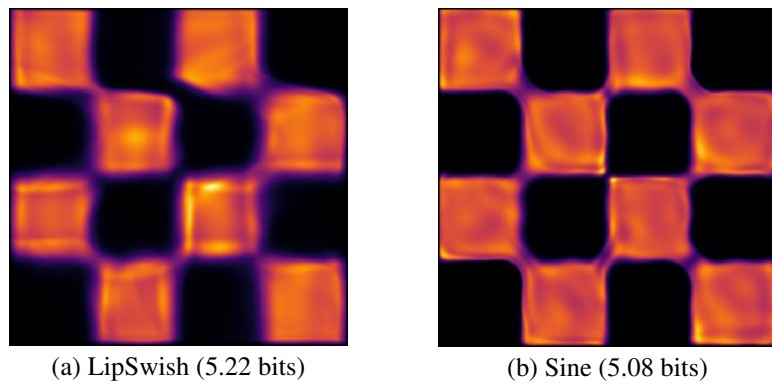

(a) LipSwish (5.22 bits)          (b) Sine (5.08 bits)

Figure 4: 8-block ResFlow with different activation function trained on Checkerboard dataset.

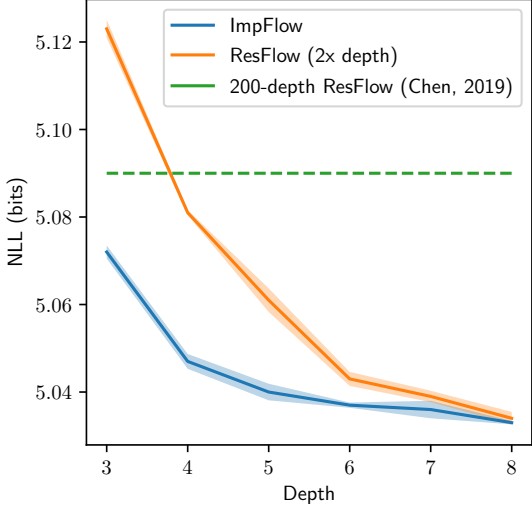

Figure 5: Test NLL (in bits) by varying the network depth. Lower is better.

## D.4 DENSITY MODELING ON TABULAR DATASETS

We use the same data preprocessing as Papamakarios et al. (2017), including the train/valid/test datasets splits. For all models, we use a batch size of 1000 (both training and testing) and learning rate of $10^{-3}$ for the Adam optimizer. The main settings are the same as Chen et al. (2019) on the toy2D dataset. The residual blocks are 4-layer MLPs with 128 hidden units. The ResFlows use 10

blocks and ImpFlows use 5 blocks to ensure the same amount of parameters. And we use a 20-block ImpFlow for a better result. Also, we use the Sine activation as $\frac{1}{2\pi}\sin(2\pi\mathbf{x})$. We do not use any ActNorm or BatchNorm layers. For the Lipschitz coefficient, we use $c = 0.9$ and the iteration error bound for spectral normalization is $10^{-3}$.

For the settings of our scalable algorithms, we use brute-force computation of the log-determinant term for POWER and GAS datasets and use the same estimation settings as Chen et al. (2019) for HEPMASS, MINIBOONE and BSDS300 datasets. In particular, for the estimation settings, we always exactly compute 2 terms in training process and 20 terms in testing process for the log-determinant series. We use a geometric distribution of $p = 0.5$ for the distribution $p(N)$ for the log-determinant term. We use a single sample of $(n, \mathbf{v})$ for the log-determinant estimators for both training and testing.

We train each expeiment on a single NVIDIA GeForce GTX 2080Ti for about 4 days for ResFlows and 6 days for ImpFlows. For 20-block ImpFlow, we train our model for about 2 weeks. However, we find that the 20-block ImpFlow will overfit the training dastaset for MINIBOONE because this dataset is quite small, so we use the early-stopping technique.

### D.5 DENSITY MODELING ON IMAGE DATASETS

For the CIFAR10 dataset, we follow the same settings and architectures as Chen et al. (2019). In particular, every convolutional residual block is

$$\text{LipSwish} \to 3 \times 3 \text{ Conv} \to \text{LipSwish} \to 1 \times 1 \text{ Conv} \to \text{LipSwish} \to 3 \times 3 \text{ Conv}.$$

The total architecture is

$$\text{Image} \to \text{LogitTransform}(\alpha) \to k \times \text{ConvBlock} \to [\text{Squeeze} \to k \times \text{ConvBlock}] \times 2,$$

where ConvBlock is i-ResBlock for ResFlows and ImpBlock for ImpBlock, and $k = 4$ for ResFlows and $k = 2$ for ImpFlows. And the first ConvBlock does not have LipSwish as pre-activation, followed as Chen et al. (2019). We use ActNorm2d after every ConvBlock. We do not use the FC layers (Chen et al., 2019). We use hidden channels as $512$. We use batch size of $64$ and the Adam optimizer of learning rate $10^{-3}$. The iteration error bound for spectral normalization is $10^{-3}$. We use $\alpha = 0.05$ for CIFAR10.

For the settings of our scalable algorithms, we use the same as Chen et al. (2019) for the log-determinant terms. In particular, we always exactly compute 10 terms in training process and 20 terms in testing process for the log-determinant series. We use a possion distribution for the distribution $p(N)$ for the log-determinant term. We use a single sample of $(n, \mathbf{v})$ for the log-determinant estimators for both training and testing.

We train each ResFlow on a single NVIDIA GeForce GTX 2080Ti and each ImpFlow on two cards of NVIDIA GeForce GTX 2080Ti for about 6 days for ResFlows and 8 days for ImpFlows. Although the amount of parameters are the same, ImpFlows need more GPU memory due to the implementation of PyTorch for the backward pass of implicit function.

For the CelebA dataset, we use exactly the same settings as the final version of ResFlows in Chen et al. (2019), except that we use the Sine activation of the form as $\frac{1}{2\pi}\sin(2\pi\mathbf{x})$.

## E IMPFLOW SAMPLES

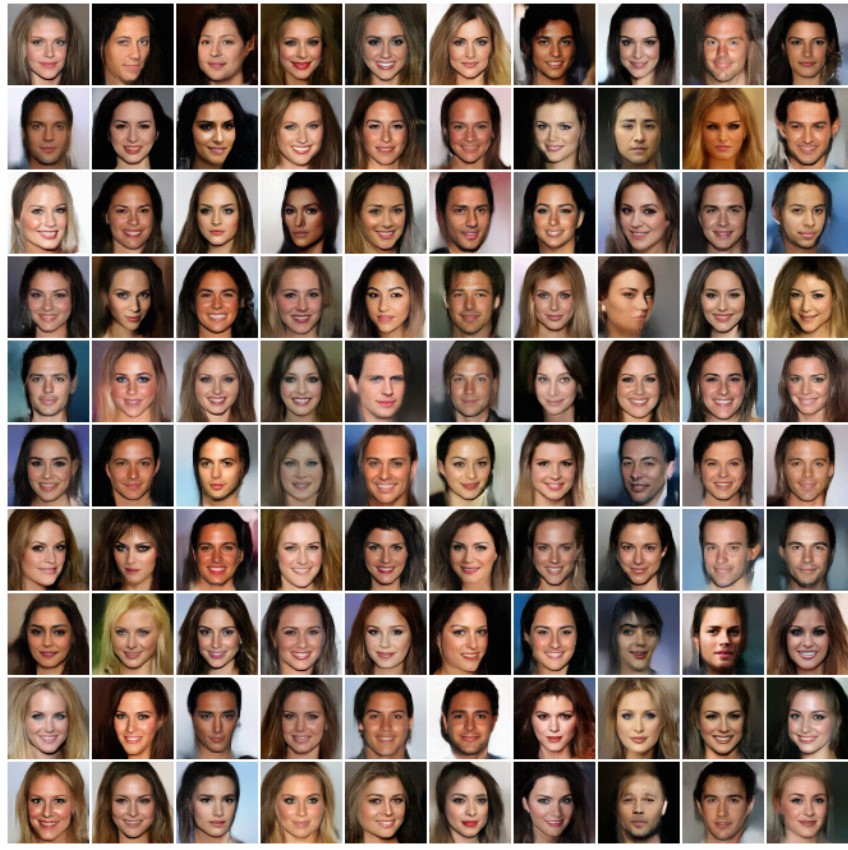

Figure 6: Qualitative samples on 5bit 64×64 CelebA by ImpFlow, with a temperature of 0.8(Kingma & Dhariwal, 2018)

