# OpenReview forum: "Implicit Normalizing Flows"
_ICLR.cc/2021/Conference — ICLR 2021 Spotlight_

### Official Review · AnonReviewer4 · 2020-10-27
**Very good submission, minor points remain open**

**Rating:** 8
**Confidence:** 4

**Review:**

# Summary
The authors concerns the question of how expressive invertible functions can be constructed. Their ansatz is the defining an invertible layer implicitly, using the root of an equation. While this approach is more general, they employ residual flows (ResFlows) to formulate a particular realisation of such an equation, calling the model ImpFlow. They show that the resulting function space is strictly richer than that of ResFlows. They further demonstrate how ImpFlows can be trained and evaluated. Empirically, ImpFlows outperform ResFlows on all considered tasks.

# Strong and weak points
## Pros:
- Implicit functions are a new way to formulate invertible functions
- The proposed formulation using ResFlows is clearly presented
- ImpFlows are strictly more expressive than ResFlows, allowing arbitrary Lipschitz constants
- Experiments highlight the improvements
- In general: very concise and well-guiding writing

## Cons:
- Implicit functions are more expensive to evaluate than explicit functions (here, +50% in execution time)

# Recommendation
The development of expressive invertible functions is key in applications that involve invertible functions, like density estimation using normalising flows. This paper proposes a novel framework to formulate such invertible expressive functions implicitly. The results are not game-changing, but consistently outperform its closest relative. Together, this is a solid work that should clearly be accepted.

# Questions
- I would like to see some analysis on how a Lipschitz function influences the Lipschitzness of the inverse. In particular, in Section 6 you could mention the relation between the "Lipschitz coefficient" c and the achievable Lipschitz constants of an ImpFlow layer. I naively would guess the bound $L < (1 + c) / (1 - c)$.
- Can you give an intuition how large the improvements on the density estimation datasets are? I checked [paperswithcode](https://paperswithcode.com/sota/density-estimation-on-uci-power) on the POWER dataset and the models are clearly outperformed by other methods. To be explicit: I don't require that every new architecture has to beat the state of the art in all possible tasks, especially when it comes with a fresh idea and thorough theory. But can you give your best guess about why other approaches achieve significantly better results?
- When comparing ImpFlow and ResFlow, I suggest that you should additionally have a variant of ResFlow with the same *execution time* in addition to *number of parameters* as a corresponding ImpFlow. I think the tradeoff involved is not only between quality and the number of parameters to store, but the also execution time of a model.

---

> ### Author Response · Authors · 2020-11-17
> **Thanks for the positive comments and acknowledgment of our contribution**
>
> We'd like to thank the reviewer for the positive comments and acknowledgment of our contribution. Below we address the reviewer's comments.
>
> #### Q1: I would like to see some analysis on how a Lipschitz function influences the Lipschitzness of the inverse.
>
> Thanks for the suggestion. We've added it in Appendix.B, Lemma 7. And the bound is indeed $L<(1+c)/(1-c)$, as suggested by the reviewer.
>
> #### Q2: About the results of tabular datasets.
>
> We've updated the results of tabular datasets and included the baselines. The original results are not strong due to the shallow architecture. Please see Table 2 of the updated version. We trained a 20-block ImpFlow on tabular datasets and achieved SOTA results on GAS and HEPMASS datasets.
>
> #### Q3:  I suggest that you should additionally have a variant of ResFlow with the same execution time in addition to the number of parameters as a corresponding ImpFlow.
>
> Thanks for the suggestion. We will add a result under the same execution time in our camera-ready version.
>
> #### Q4: Typos.
>
> Thanks for the suggestion. we've fixed the typos.

---

### Official Review · AnonReviewer2 · 2020-10-27
**Overall a nice work!**

**Rating:** 7
**Confidence:** 3

**Review:**

# Summary
This work is about a new architecture for normalizing flows inspired by implicit neural networks. In particular, the authors show that a specific implicit neural network build from residual blocks defines a bijective map. From this insight, they show how to efficiently train such architecture by estimating the log Jacobian and solving the inverse problem related to the implicit architecture. On top of this achievement, the authors thoroughly study the attractiveness of their architecture compared to i-res-flow. They demonstrate analytically and empirically that implicit normalizing flows are strictly more expressive than i-res-flow.

# Major comments
## Pros:
Overall the paper is well written and pleasant to read. The idea is novel and is well introduced. Moreover, the paper provides a theoretical justification for the expressivity gain made by the newly introduced architecture with respect to I-res-flow. The experimental results are good although not defining a new standard in density estimation tasks.

## Cons:
1) As I said the paper is well written overall. However, I took quite some time to fully understand section 4 (in particular 4.2), I think this section should be clarified. After taking the time, everything appears convincing but I think the flow could be improved. I checked the proofs in 4.2 and everything seems correct, great job! You should mention that everything is proved in the appendix, I was not sure it was the case the first time I read it.
2) Regarding the experiments I have two concerns. First, you provide only results about one specific value of L without stating how you chose it. It feels like you selected it to show the performance gain brought by your method, it may be fine but you should at least mention it clearly. Else you should perform a full architecture search both for resflow and for impflow and show the best results of each architecture. My second concern is about table 2, I think you should mention other sota NF architectures (such as the ones based on monotonic transformation) and maybe discuss a possible explanation for the large gap.

## What could be done to address my comments
1)  You could maybe directly state that $\mathcal{R}_2 \subset_\neq \mathcal{F}_2 = \mathcal{I}$ instead of corollary 1.
2) Clarify how you selected the number of steps. I saw you mention that you copy the experimental setting of Chen et al. 2019. However, they use 200 steps for the 2D grid and do not provide results for the UCI dataset (or I missed it maybe).

# Minor comments
p3: "inverible".
4.1: $J_f$ inconsistent notation wrt (1).
def 2: missing . at the end of the equation.
"2-norm": do you mean $L_2$ norm?
"in 1-D input" -> "In the 1D input"
D.4: "Tablular"

---

> ### Author Response · Authors · 2020-11-17
> **Thanks for the positive comments and acknowledgment of our contribution**
>
> We'd like to thank the reviewer for the positive comments and acknowledgment of our contribution. Below we address the reviewer's comments.
>
> #### Q1: You should mention that everything is proved in the appendix
>
> Thanks for the suggestion. We've mentioned it at the beginning of Sec.4.2. in our updated version.
>
> #### Q2: Clarify how you selected the number of steps (L).
>
> Thanks for the suggestion. We include a new experiment varying the depth (steps). ImpFlows are always better than ResFlows and require fewer blocks than ResFlow to achieve the same likelihood. Please refer to Appendix.D.3 and Figure 5 in our updated version.
>
> #### Q3: You should mention other sota NF architectures in Table 2.
>
> We've updated the results of tabular datasets and included the baselines. The original results are not strong due to the shallow architecture. Please see Table 2 of the updated version. We trained a 20-block ImpFlow on tabular datasets and achieved SOTA results on GAS and HEPMASS datasets.
>
> #### Q4: You could maybe change the statement of Corollary 1.
>
> Thanks for the suggestion. We've changed the statement of Thm.2 and Cor.1 to use the notation of $\mathcal{F}_2$.
>
> #### Q5: Typos.
> Thanks for the suggestion. We've fixed the typos in our updated version.

---

> > ### Comment · AnonReviewer2 · 2020-11-23
> > **Thanks for taking my comments into account.**
> >
> > I would like to thank the authors for their response. The additional results seem to strengthen further the paper, congrats'!

---

### Official Review · AnonReviewer3 · 2020-10-28
**Good idea, some important details missing.**

**Rating:** 7
**Confidence:** 4

**Review:**

Summary: The paper introduces an invertible transformation implicitly via the roots of an equation. As a result, they claim that this implicit transformation is in theory a superset of Residual Flows. Further, they show improved performance empirically in NLL on CIFAR10 in varying settings.

Review:
The paper introduces a novel an interesting method to parametrize invertible transformations via an implicit. The method in its current form can actually be interpreted as two Residual Blocks, but where the inverse parametrisation of the second block is used in the forward direction (as the authors show in Eq. 6). Nevertheless, I find the proposed perspective interesting and a promising future direction. The authors give detailed proofs of the flexibility of their method. The authors also propose a more memory efficient solution for training.

Weaknesses: Experimentally there are some unanswered questions.
- As Implicit flows require iterative solutions during training, they will most likely be slower. Even if ImpFlows are slower, it is really necessary to clearly highlight this difference in computation time. Further, ImpFlows are probably even faster during sampling as half of the Residual Blocks are now aligned with the sampling direction.
- The new gradient computation from Eq (16) is only very briefly discussed in the paper. Since the authors claim gains in memory efficiency, these results need to be backed up empirically.
- Thm 1 can also be derived from the perspective of two stacked residual blocks (second inverted). It would help readers to already introduce that perspective here.

If the authors address the points raised in Weaknesses, I will consider raising my score.

Minor comments:
Typos. The paper has some typos and in many cases a spell checker could correct these. For instance: Inverible (Sec 3.2) Esitamated (algorithm 1.).
Rephrasing (sec. 4.2). "Note that we define R_1 = R" -> "By definition of Eq. 4 and Eq. 5, R_1 = R"
The overload in Algorithm 1 is somewhat difficult to parse, consider changing "g(z) = F (z, x; theta)"

After rebuttal:
I am satisfied with the reply of the authors and I have raised my score to 7.

---

> ### Author Response · Authors · 2020-11-17
> **Thanks for the positive comments and acknowledgement to our novelty and significance**
>
> We'd like to thank the reviewer for the positive comments and acknowledgement to our novelty and significance. Below we address the reviewer's comments.
>
> #### Q1:  It is really necessary to clearly highlight this difference in computation time.
> Thanks for the suggestion. We decomposed the single-batch computation costs for ResFlow and ImpFlow in our updated paper. There are three different settings: training, inference (density computation), and sampling. The training time of ImpFlow (4.152s) is ~40% longer than ResFlow (2.910s). However, the inference time gap is much smaller (2.905s vs 2.656s), since ImpFlow's additional overhead on fixed-point solver (0.445s) is marginal, relative to the overhead of approximating the log determinant (2.370s).
>
> Moreover, ImpFlow's sampling time is ~40% FASTER than ResFlow, as half of the blocks are already aligned in the sampling direction. Fast sampling is particularly desirable since it is the main advantage of flow-based models over autoregressive models. Please refer to Appendix.C.2, Table 4, and Table 5 for details.
>
> #### Q2: Why Eqn. (16) is memory efficient?
> We suspect there could be some misunderstandings. By "memory efficient", we mean that our algorithm does not need to store the intermediate results of the Broyden's iterations. We've changed our discussion under Eqn.(16) by adding "the quasi-Newton iterations of the forward pass" to clarify the statement.
>
> Details: In the forward pass, ImpFlows need to solve the fixed-point by the quasi-Newton iterations. Every iteration is like "$z^{[i+1]}=G^{[i]}(z^{[i]}; \theta)$", and after $L$ steps we get $z^* \approx z^{[L]}$. In the backward pass, we need to compute the loss gradient w.r.t. $\theta$. By backpropagation, we firstly have the loss gradient w.r.t $z^{[L]}$. If we do not use Eqn.(16), we need to backpropagate along $z^{[L]},z^{[L-1]}\cdots, z^{[1]}$ to compute the gradient w.r.t. $\theta$. Therefore, we need to save all the $L$ intermediate results $z^{[1]},\cdots, z^{[L]}$ during the forward pass. This costs $O(L)$ GPU memory. Instead, if we use Eqn.(16), we only need to save $z^{[L]}$ during the forward pass, and this costs only $O(1)$ GPU memory, which is independent of the iteration steps in the forward pass and treats the root solvers (such as the Broyden's method) as a black-box.
>
> #### Q3: Thm. 1 can also be derived from the perspective of two stacked residual blocks (second inverted)
> Thanks for the suggestion. We've added this discussion under Thm. 1 in our updated version.
>
> #### Q4: Typos and statement of Algorithm 1.
> Thanks for the suggestion. We've fixed the typos and changed the statement of Algorithm 1 in the updated version.

---

### Official Review · AnonReviewer1 · 2020-10-29

**Rating:** 8
**Confidence:** 4

**Review:**

Paper summary:

The authors propose ImpFlow, an implicit normalizing flow defined around solving the equation x + g_x(x) = z + g_z(z) where g_x and g_z have Lipschitz < 1.

Solving z given x, or x given z, gives the forward and inverse passes of ImpFlow. Both directions require solving a root finding problem, hence this model is implicit in both directions, contrasting with prior works (e.g. ResFlow) where at least one direction is always explicit.

ImpFlow is equivalently a composition of ResFlow and the inverse of a ResFlow.

In the context of expressiveness, the forward pass a ResFlow is (1 + L)-Lipschitz while, importantly, the inverse is (1 / (1 - L))-Lipschitz, for some L < 1. Since ImpFlow makes use of the inverse of ResFlows in its construction, it can model arbitrary Lipschitz transformations.

In the context of likelihood evaluation and training, the estimators from ResFlows can be used and the main difference is the use of implicit function theorem to differentiate through the root finding procedure. Authors claim the total compute cost is comparable, though solving the root finding problem does seem to introduce some overhead, going from 3.189s to 4.462s (~40% increase) for each iteration of training on CIFAR-10.

Strong points:

The use of the inverse of ResFlows in both directions of a normalizing flow is interesting, and result of having higher Lipschitz is convincing.

In addition to the well-written series of lemmas and theorems, the additional capacity of ImpFlow over ResFlow is nicely illustrated on a 1D example.

Experiments indicate ImpFlow is slightly more performant in log-likelihood than ResFlows for the same number of parameters.

Weak points:

Is being implicit really useful? Outside of the Lipschitz requirement in this particular formulation of ImpFlows using the inverse of ResFlows, it's not clear whether having implicit formulations in both directions of a normalizing flow is advantageous. In addition to the use of a general title and method name advocating for implicit formulations, the authors do claim that "bijections with explicit forward mapping only covers a fraction of the broad class of invertible functions suggested by the first requirement, which may limit the model capacity" in the introduction. This ideally should be accompanied by a reference or rephrased to be a conjecture.

The experiments do not compare to other existing normalizing flows outside of ResFlows. The numbers reported in Table 2 (for tabular experiments) are worse than the baselines reported in Table 1 of Papamakarios et al. "Masked Autoregressive Flow for Density Estimation" while the results on CIFAR10 (Table 3) are closer to Real NVP (2016) than models like Glow or FFJORD (2018). This is likely simply due to the use of smaller models than those used previously, but if so, perhaps the authors could comment a bit more on the problems for scalability of ImpFlows. A ~40% additional time cost is mentioned once when compared to ResFlows, and an extra usage of GPU memory cost is only mentioned in the Appendix. The other parts of the paper are rather well done, but I'd liked to have seen explicit diagnostics of ImpFlows. For instance:
  - How many function evaluations were used for solving the implicit formulation?
  - Are results sensitive to the value of epsilon (for convergence of fixed point) hyperparameters? (A problem for implicit formulations is controlling the numerical error. Is this a problem for ImpFlows?)
  - Since the Lipschitz of ResFlows grows exponentially, does the empirical difference between ImpFlow and ResFlow shrink as model sizes increase?

Since the differences between ResFlow and ImpFlow are rather small (e.g. 0.01~0.02 bits/dim on CIFAR10), it'd be good to have some standard deviation across random seeds for all experiments.

Additional comments:

The citation under Lemma 1, "Chen et al. (2019, Lemma 2)" should probably be "Behrmann et al. (2019, Lemma 2)". (I assume the authors meant to refer to the bi-Lipschitz bounds derived in Behrmann et al.?)

Post-discussion:

I thank the authors for responding swiftly to all of my main concerns and for providing additional experimental results. I'm happy that authors have promised standard deviations and comparisons to ResFlow in the newly updated experiments for the camera-ready version, and I have adjusted my rating accordingly. I think this paper makes a very solid contribution to the normalizing flows literature.

---

> ### Author Response · Authors · 2020-11-17
> **Thanks for the positive comments and acknowledgment of our contribution**
>
> We'd like to thank the reviewer for the positive comments and acknowledgment of our contribution. Below we address the reviewer's comments
>
> #### Q1: Is being implicit really useful?
>
> The implicit formulation is useful. Its relaxed Lipschitz constraint has real empirical improvements, leading to SOTA results on some tabular datasets (please refer to our updated paper). For instance, on the HEPMASS dataset, our 20-block ImpFlow achieves a SOTA log-likelihood (-13.95), which improves considerably than the previous SOTA (-15.09) for normalizing flow models, to the best of our knowledge.
>
> Thanks for the suggestion on writing. We will rephrase the claims in the final version. In fact, our ImpFlow itself is a bijection that cannot be represented by an explicit forward mapping of neural networks. Furthermore, recently implicit methods have shown great promise in deep learning. They achieve more modeling flexibility by incorporating logical reasoning [1] and also improved memory consumption [2]. While exploring more implicit bijections is beyond the scope of this paper, these positive results strengthen our belief that implicit flows are a promising direction to research.
>
> #### Q2: The experiments do not compare to other existing normalizing flows outside of ResFlows.
>
> We've updated the results of tabular datasets and included the mentioned baselines. The original results are not strong due to the shallow architecture. Please see Table 2 of the updated version. We trained a 20-block ImpFlow on tabular datasets and achieved SOTA results on GAS and HEPMASS datasets.
>
> #### Q3: How many function evaluations were used for solving the implicit formulation?
> We evaluated the single-batch computation costs (running time and number of function evaluations) for ResFlow and ImpFlow in our updated version. The number of function evaluations for forward and backward are 10.7 and 22.7, respectively. Please refer to Table 4, and Table 5 in Appendix.C.2 for details.
>
> #### Q4: Are results sensitive to the value of epsilon (for convergence of fixed point) hyperparameters?
>
> Empirically, our model is not sensitive to $\epsilon_f$ in a fair range ($10^{-8}$ to $10^{-2}$). Please refer to Appendix.C.3 in our updated version.
>
> #### Q5: Since the Lipschitz of ResFlows grows exponentially, does the empirical difference between ImpFlow and ResFlow shrink as model sizes increase?
>
> We include a new experiment varying the depth. ImpFlows are always better than ResFlows. The performance gain becomes smaller as the model grows deeper since they all converge to a performance upper bound (which is related to the network structure, activation functions, and hyperparameters). Nevertheless, ImpFlow requires fewer blocks than ResFlow to achieve the same likelihood. Please refer to Appendix.D.3 and Figure 5 in our updated version.
>
> #### Q6: It'd be good to have some standard deviation across random seeds for all experiments.
>
> Thanks for the suggestion. Due to the limited time of the discussion period, we will add it in our camera-ready version.
>
> #### Q7: Typos.
>
>
> Thanks for pointing it out. We've fixed the typos in our updated version.
>
> =====================================================================
>
> [1]. Po-Wei Wang, Priya Donti, Bryan Wilder, and Zico Kolter. Satnet: Bridging deep learning and logical reasoning using a differentiable satisfiability solver. In International Conference on Machine Learning, pp. 6545–6554, 2019.
>
>
> [2]. Shaojie Bai, J. Zico Kolter, and Vladlen Koltun. Deep equilibrium models. In Advances in Neural Information Processing Systems (NeurIPS), 2019.

---

> > ### Comment · AnonReviewer1 · 2020-11-19
> > **Thanks for the prompt updates**
> >
> > Thanks for the prompt response and updates! The updates have answered some of my questions, though a couple remain. My responses below:
> >
> > First, this isn't a criticism of the method or experimentation, just a continuation of the discussion:
> >
> > ### Re: implicit being useful.
> >
> > What I meant to convey is that being non-implicit doesn't automatically imply a bounded Lipschitz. This is only true for the ResFlow parameterization. I simply think it's a good idea to separately consider implicit functions and Lipschitz constraints. For instance, I disagree with this statement:
> >
> > > In fact, our ImpFlow itself is a bijection that cannot be represented by an explicit forward mapping of neural networks.
> >
> > An implicit formulation still must model a continuous mapping (otherwise it's not differentiable), and a neural network is a universal approximator for continuous mappings. And in light of works on proving normalizing flows to be universal density approximators (e.g. NAF), it'd be difficult to say ImpFlow is theoretically better than normalizing flow models other than ResFlow.
> >
> > There are also many normalizing flows that make use of implicit formulations without a fixed Lipschitz bound. For instance, autoregressive flows don't have an analytical solution for the inverse; it needs to either invert one variable at a time or make use of a fixed point solver (e.g. MintNet). Continuous normalizing flows do not have explicit transformations in either direction, and the Lipschitz is unbounded because the value of the differential equation is unbounded.
> >
> > While I also think implicit formulations are very interesting, I don't think these models should be considered good because of the implicit formulation; instead, I see the implicit formulation as a side effect (in this case, of increasing the Lipschitz bound for ResFlow-like models (great!)), but on its own, could even be a downside. In any case, I really like the construction of ImpFlows and simply wanted to express my minor disagreement with the authors' choice of method name and statements such as the above.
> >
> > -----
> >
> > Regarding the newly added experiments,
> >
> > > We've updated the results of tabular datasets and included the mentioned baselines.
> >
> > Great! But it seems the difference between L=5 and L=20 is quite large. Given this, the comparison to ResFlow is now not so clear and seems to only be carried out in the shallow regime. That is, it's not clear if this gain in performance is due to a higher practical Lipschitz bound, or can be followed suit by a ResFlow. Is it possible to have comparable ResFlows to complement the deeper ImpFlows?
> >
> > > The number of function evaluations for forward and backward are 10.7 and 22.7.
> >
> > Forgive me if I have my criticism hat on, but why are these values (and the computation time) computed using a model with c = 0.5? This is a much smaller value than used in the main paper experiments. With c = 0.5, the inverse of ResFlow has a Lipschitz bound of 1 / (1 - c) = 2, which is the same as the forward pass of ResFlow. So a L-block ImpFlow with c=0.5 can no longer model larger Lipschitz constants than a 2L-block ResFlow.
> >
> > This seems quite important because the number of fixed point iterations should empirically depend quite heavily on the Lipschitz constants of g_x and g_z (which are bounded by c), especially for the first few iterations of Broyden's method before it gets a good guess of the Jacobian inverse. For instance, Banach's fixed-point theorem gives convergence rates in terms of the Lipschitz constant; the smaller the faster the convergence.
> >
> > If the authors want to show estimates of compute costs, I'd liked to have seen compute costs that reflect the same setting as models shown in the main text, or even better, an analysis of how the compute cost scales with c.
> >
> > ### Sampling costs.
> >
> > Just to be clear, are you also using the same procedure to invert ResFlows as the ImpFlows? (i.e. Broyden's method, same convergence criteria) Because ResFlows used a simpler fixed point (no pseudo-Newton, no line search) in their paper. On this note, do the authors believe Broyden's method is helping in reducing the number of network evaluations?
> >
> > > Our model is not sensitive to $\epsilon_f$
> >
> > > Please refer to Appendix.D.3 and Figure 5 in our updated version.
> >
> > Great, both of these look good to me. I appreciate the authors completing these experiments during the discussion period.
> >
> > > Due to the limited time of the discussion period, we will add [standard deviation across random seeds] in our camera-ready version.
> >
> > Thanks. This does seem quite important, as it would give us a better sense of how important and reliable increasing the Lipschitz bound is for these data sets.

---

> > > ### Author Response · Authors · 2020-11-23
> > > **Thanks for the suggestions!**
> > >
> > > We'd like to thank the reviewer for the interest in our work and the positive updated comments. Below we address the reviewer's comments.
> > >
> > > #### Q1: About the implicit formulation.
> > > Thanks for the constructive discussion. In our initial version, we choose to term our method as the more general "implicit normalizing flows" because we think the implicit view itself has some insights. Though we mainly discussed the implicit version of residual flows, we conjecture the general implicit bijection family has more to explore. We understand and agree with the reviewer's point, and will rephrase our statements more rigorously in our camera-ready version.
> > >
> > > #### Q2: Comparable ResFlows with deeper networks.
> > > Thanks for the suggestion. We will add it to our camera-ready version due to the time limitation of the discussion period.
> > >
> > > #### Q3: About the time costs and function evaluation numbers of ImpFlows.
> > > Thanks for the suggestion. We've added a full table of c=0.5,0.6,0.7,0.8 and 0.9 to compare the computation costs between ImpFlows and ResFlows, as shown in the updated version of Table 5 and Table 6. Although the computation overhead of ImpFlows increases as $c$ becomes large, the main conclusion remains the same: The training time of ResFlows is still comparable with ResFlows (40%\~80%). However, the inference time gap is much smaller (10%\~20%), since ImpFlow's additional overhead of fixed-point solver is marginal, relative to the overhead of approximating the log determinant. And the sampling time of ImpFlows is faster (40%\~60%) than that of ResFlows, as half of the blocks are already aligned in the sampling direction. Fast sampling is particularly desirable since it is the main advantage of flow-based models over autoregressive models. For the function evaluation numbers, we've corrected the number we reported for $c=0.5$, and the numbers of function evaluations for forward and backward are 8.2 and 13.5. And for $c=0.9$, the numbers are 13.9 and 28.4, respectively. This result is consistent with the reviewer's points. However, we note that this does not affect the conclusion, because this overhead is comparable.
> > >
> > > #### Q4: About the sampling method.
> > > Thanks for pointing it out. We use the same sampling code as the released version of ResFlows, for the fair comparison. We've made it more clear in our updated version in Appendix C.2.
> > >
> > > #### Q5: Do the authors believe Broyden's method is helping in reducing the number of network evaluations?
> > > This is a very good question. Empirically, we find that the fixed-point iteration method needs more number of function evaluations, but for some big $\epsilon_f$, it sometimes costs less time than the Broyden's method (due to the cost for approximating the inverse Jacobian). However, during the training phase, using Broyden's method is helpful for the precision of the fixed-point solution because we can set a small $\epsilon_f$. And the fixed-point iteration method sometimes cannot achieve such precision and may cost much more time and the number of function evaluations. However, for the sampling phase, we do not need a high precision solution because the image data is discrete. So we simply use fixed-point iterations with a little bigger tolerance.

---

### Decision · Program_Chairs · 2021-01-07
**Final Decision**

**Decision:**

Accept (Spotlight)

**Comment:**

This is a clear accept. Solid and timely work extending normalizing flows to implicitly defined mappings. Convincing presentation. Supported by all four reviewers. Best paper in my batch. Has the potential to spark further developments in the field. I recommend to feature this paper as a spotlight.